# Disease progression modelling reveals heterogeneity in trajectories of Lewy-type α-synuclein pathology

Sophie E. Mastenbroek [1,2,3] ✉, Jacob W. Vogel [4], Lyduine E. Collij [1,2,3], Geidy E. Serrano[5], Cécilia Tremblay[5], Alexandra L. Young [6,7], Richard A. Arce[5], Holly A. Shill[8], Erika D. Driver-Dunckley[9], Shyamal H. Mehta[9], Christine M. Belden[5], Alireza Atri[5,10], Parichita Choudhury[5], Frederik Barkhof [1,2,11], Charles H. Adler[9], Rik Ossenkoppele [3,12,13], Thomas G. Beach[5] & Oskar Hansson [3,14] ✉

Lewy body (LB) diseases, characterized by the aggregation of misfolded α-synuclein proteins, exhibit notable clinical heterogeneity. This may be due to variations in accumulation patterns of LB neuropathology. Here we apply a data-driven disease progression model to regional neuropathological LB density scores from 814 brain donors with Lewy pathology. We describe three inferred trajectories of LB pathology that are characterized by differing clinicopathological presentation and longitudinal antemortem clinical progression. Most donors (81.9%) show earliest pathology in the olfactory bulb, followed by accumulation in either limbic (60.8%) or brainstem (21.1%) regions. The remaining donors (18.1%) initially exhibit abnormalities in brainstem regions. Early limbic pathology is associated with Alzheimer's disease-associated characteristics while early brainstem pathology is associated with progressive motor impairment and substantial LB pathology outside of the brain. Our data provides evidence for heterogeneity in the temporal spread of LB pathology, possibly explaining some of the clinical disparities observed in Lewy body disease.

The aggregation of misfolded α-synuclein is a pathologic hallmark of Lewy body (LB) disease including Parkinson's disease (PD) and dementia with Lewy bodies (DLB)[1]. LB diseases have a long prodromal phase where central nervous system (CNS) or peripheral nervous system (PNS) clinical signs and symptoms may be present, such as rapid eye movement sleep behavior disorder, hyposmia, depression, and delayed gastric emptying and constipation[2]. Despite having a common underlying pathophysiology, LB diseases are associated with

[1]Department of Radiology and Nuclear Medicine, Vrije Universiteit Amsterdam, Amsterdam University Medical Center, location VUmc, Amsterdam, The Netherlands. [2]Amsterdam Neuroscience, Brain imaging, Amsterdam, The Netherlands. [3]Clinical Memory Research Unit, Department of Clinical Sciences Malmö, Faculty of Medicine, Lund University, Lund, Sweden. [4]Department of Clinical Sciences Malmö, Faculty of Medicine, SciLifeLab, Lund University, Lund, Sweden. [5]Banner Sun Health Research Institute, Sun City, AZ, USA. [6]Department of Neuroimaging, Institute of Psychiatry, Psychology and Neuroscience, King's College London, London, UK. [7]Centre for Medical Image Computing, Department of Computer Science, University College London, London, UK. [8]Department of Neurology, Barrow Neurological Institute, Phoenix, AZ, USA. [9]Department of Neurology, Parkinson's Disease and Movement Disorders Center, Mayo Clinic, Scottsdale, AZ, USA. [10]Department of Neurology, Center for Mind/Brain Medicine, Brigham & Women's Hospital & Harvard Medical School, Boston, MA, USA. [11]Institutes of Neurology & Healthcare Engineering, University College London, London, UK. [12]Alzheimer Center Amsterdam, Neurology, Vrije Universiteit Amsterdam, Amsterdam University Medical Center location VUmc, Amsterdam, The Netherlands. [13]Amsterdam Neuroscience, Neurodegeneration, Amsterdam, The Netherlands. [14]Memory Clinic, Skåne University Hospital, Malmö, Sweden. ✉e-mail: s.e.mastenbroek@amsterdamumc.nl; oskar.hansson@med.lu.se

varying clinical syndromes, with PD initially presenting with parkinsonian motor symptoms[3] and DLB with cognitive impairment and dementia. By international consensus, the diagnosis of DLB is assigned if dementia is diagnosed prior to or within 1 year of the first signs of parkinsonism[4]. Furthermore, symptom heterogeneity within each disease entity has also been reported[5–9]. These and other clinical variations may be attributed to differences in the extent and spatial distribution of the underlying LB pathology[10–13]. However, the trajectories of LB accumulation and the degree of variation herein remains incompletely understood.

Significant efforts have been made to characterize the pathological progression of LB diseases, leading to the development of several distinct staging systems. Amidst these, the Braak staging scheme is among the most widely recognized[14]. It postulates that LB pathology starts simultaneously in lower brainstem regions and the olfactory bulb, possibly originating from the periphery and penetrating into the nervous system through the olfactory and intestinal epithelium[15] (Fig. 1A). Next, pathology spreads sequentially from the brainstem to the mesencephalon, limbic structures, and the cortex, while olfactory bulb pathology does not spread beyond non-olfactory structures. However, this hypothesis has been criticized in the literature[16,17] because it fails to account for alternative patterns of LB pathology. Hence, other models have been devised proposing that LB pathology may instead solely originate in the CNS and propagate along more than one route. For instance, the Unified Staging System for Lewy Body Disorders (USSLB) posits that, in most cases, LB pathology starts in the olfactory bulb[18] (Fig. 1B). From there, it diverges into a brainstem-predominant or limbic-predominant pathway, ultimately spreading to cortical regions. More recently, a third scenario was proposed, suggesting that both frameworks might be true by representing two distinct subtypes of LB diseases[19,20]. In this brain-first vs. body-first hypothesis, LB diseases may comprise two distinct subtypes, with one originating from the periphery (i.e., body-first) and one originating from the CNS (i.e., brain-first), possibly explaining some of the clinical heterogeneity that has been reported (Fig. 1C). To date, there is no consensus on which staging system describes the pathological progression of LB diseases most accurately.

One methodological approach to elucidate disease trajectories of LB diseases in greater depth is the implementation of data-driven disease progression modeling. Whilst longitudinal data is required to examine the temporal spread of pathology within a single individual, cross-sectional data can be used to generate hypotheses of common patterns of the temporal spread of pathology within a population. This approach assumes that individuals represent different stages (and possibly subtypes) of a common disease progression pattern. This procedure has long been used by neuropathologists to develop disease staging systems. However, recent advances in machine learning algorithms are allowing more complex hypotheses to be tested that involve much larger numbers of regions, evaluate the evidence for disease subtypes, and appropriately handle statistical uncertainty[13]. One such probabilistic algorithm is the Subtype and Stage Inference (SuStaIn) model, which combines disease progression modeling with clustering to identify groups of individuals with distinct spatiotemporal disease trajectories, using heterogeneous cross-sectional data[21]. Recently, a new implementation of SuStaIn was developed (Ordinal SuStaIn), enabling the algorithm to be applied to semi-quantitative pathological data[22]. This new version of SuStaIn was able to accurately capture heterogeneity in disease trajectories of post-mortem TAR DNA-binding protein 43 (TDP-43) data and provided biologically plausible subtypes[23].

Here we applied ordinal SuStaIn to 814 autopsy cases with evidence of LB pathology to examine the progression and heterogeneity of LB accumulation across 10 brain regions. Next, we characterized subgroups by assessing differences regarding demographics, pathological data, and clinical symptoms. Finally, we investigated hypothetical differences in trajectories of peripheral LB pathology.

## Results

### Subjects

The SuStaIn modeling cohort comprised 814 clinicopathologically characterized subjects from the Arizona Study of Aging and Neurodegenerative Disorders (AZSAND)/Brain and Body Donation Program (BBDP)[24]. AZSAND/BBDP recruitment is directed at cognitively normal elderly and subjects with a clinical diagnosis of Alzheimer's disease (AD), PD, or cancer. Inclusion criteria are the absence of a hazardous infectious disease and consent to annual clinical assessments and autopsy at the time of death. In the current study, we included all autopsied BBDP subjects who showed evidence for LB pathology and had at least 7 brain regions assessed for Lewy-type α-synuclein (LTS) density. Demographic information can be found in Table S1. Mean age at death was 81.8 ± 7.7 years, 38.9% were females, 41.8% were APOE-ε4 carriers, and total brain LTS pathology (0–4 for each of 10 regions with range 0–40[25]) was on average 18.1. In addition, 285 (35%) had a clinicopathologic diagnosis of AD, 168 (20.6%) PD, 19 (2.3%) DLB, 78 (9.6%) mixed AD and PD, 141 (17.3%) mixed AD and DLB, 90 (11.1%) incidental Lewy body disease (ILBD), and 33 (4.1%) any other diagnosis (Table S2).

### Three heterogeneous disease progression patterns of Lewy-type α-synucleinopathies

The ordinal implementation of the SuStaIn algorithm was applied to neuropathological LTS density scores (0–4, with 0 = no and 4 = very severe pathology) assessed in 10 brain regions (olfactory bulb, three limbic, three brainstem, and three neocortical). Model-fit statistics of 10-fold cross-validation implied that three spatiotemporal subtypes of misfolded α-synuclein aggregation best supported the data (Fig. S1). Based on the ordering of brain regional severity, the subtypes were termed "S1: OBT-early/Limbic-early", "S2: OBT-early/Brainstem-early", and "S3: Brainstem-early/OBT-later".

In the S1 (OBT-early/Limbic-early) subtype, the earliest involvement was observed in the olfactory bulb and tract (OBT) and limbic regions, in particular the amygdala, followed by brainstem regions and culminating in neocortical regions (Fig. 2A). The S2 (OBT-early/Brainstem-early) subtype was inferred to start identically with earliest involvement of the OBT, followed, however, by brainstem rather than limbic regions (Fig. 2B). After the brainstem, subsequent involvement

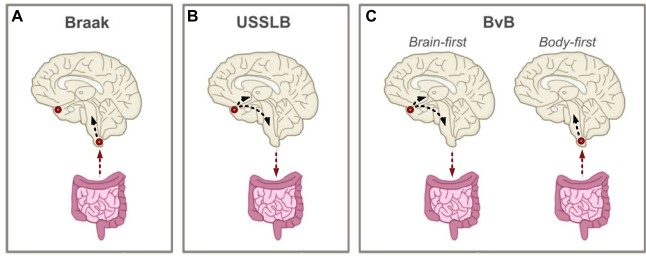

**Fig. 1 | Hypothetical disease progression models of Lewy body pathology.** Summary of theoretical disease models describing the progression of Lewy body (LB) pathology. Origins of LB pathology are designated with red circles. Arrows denote direction of spreading. **A** The Braak staging scheme postulates that LB pathology enters the brain through the gut and/or the nasal epithelium, resulting in simultaneous deposition in brainstem regions and the olfactory bulb[14,15]. Only brainstem pathology subsequently propagates throughout the brain. **B** The Unified Staging System for Lewy Body Disorders (USSLB) posits that LB pathology starts in the olfactory bulb in most cases, followed by either brainstem or limbic regions[18]. It states that pathology always starts in the brain and propagates to the body. **C** The brain-first vs. body-first (BvB) model hypothesizes the existence of two subtypes of LB diseases, with one originating from the brain (i.e., brain-first) and the other from the body (i.e., body-first)[19,53]. In the former, pathology starts in olfactory regions, followed by either brainstem or limbic regions. In the latter, pathology enters the brain through brainstem regions.

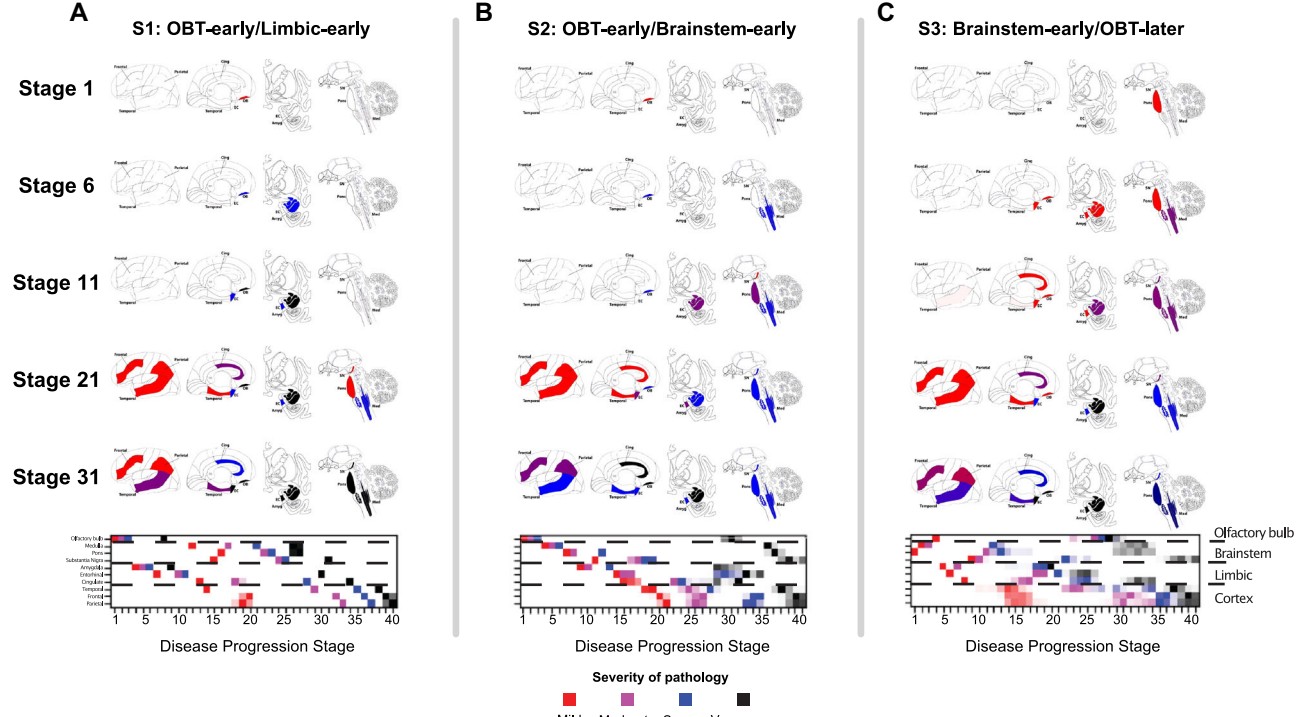

**Fig. 2 | Three spatiotemporal trajectories of LB pathology.** SuStaIn-inferred disease progression patterns of regional Lewy-type α-synuclein pathology. Three distinct trajectories were identified and termed (**A**) "S1: OBT-early/Limbic-early", (**B**) "S2: OBT-early/Brainstem-early", and (**C**) "S3: Brainstem-early/OBT-later". Brain maps of α-synuclein deposition are shown for SuStaIn stage 1, 6, 11, 21, and 31, with colors indicating severity of pathology (red = mild, purple = moderate, blue = severe, black = very severe). Below, positional variance diagrams are shown. Each box represents the certainty that a brain region has reached a certain level of pathology (mild, moderate, severe, and very severe) at a given SuStaIn (i.e., disease progression) stage, with darker colors representing more confidence. Brain schematics were generated using: https://github.com/AllenInstitute/hba_brain_schematic (Copyright © 2023. Allen Institute. All rights reserved.). Amyg amygdala, Cing anterior cingulate, EC entorhinal cortex, Med medulla, SN substantia nigra, SuStaIn Subtype and Stage Inference.

of limbic, and neocortical regions was observed. In contrast to the other two subtypes, the S3 (Brainstem-early/OBT-later) subtype demonstrated substantial OBT pathology only in later SuStaIn stages (>25) (Fig. 2C). Instead, brainstem regions, especially the pons and medulla, showed heavier early engagement, progressing thereafter to limbic and neocortical regions.

When including only those subjects with complete LTS data (i.e., measured in all 10 brain regions), three highly similar subtypes were observed (Fig. S2), demonstrating robustness of the results.

The OBT is a small and delicate region that is prone to inadequate sampling and underestimation of pathology. To confirm that the S3 (Brainstem-early/OBT-later) subtype was not driven by false-negative OBT cases, we performed a sensitivity analysis in which we re-stained and re-examined all OBT-negative samples (n = 40) from the S3 (Brainstem-early/OBT-later) subtype. Four cases were deemed inadequate and five cases were reclassified as OBT-positive, ranging from moderate to very severe pathology. Running SuStaIn on the adapted dataset yielded three subtypes with comparable disease trajectories (Fig. S3).

Comparison of the average regional LB density between subtypes revealed that subjects in the S1 (OBT-early/Limbic-early) subtype had more pathology in the OBT and amygdala compared to the other subtypes, and more pathology in the transentorhinal cortex compared to S2 (OBT-early/Brainstem-early) (Fig. 3A). In contrast, both the S2 (OBT-early/Brainstem-early) and S3 (Brainstem-early/OBT-later) subtypes had more pathology in the pons and medulla than the S1 (OBT-early/Limbic-early) subtype. Finally, the S3 (Brainstem-early/OBT-later) subtype had less pathology in the OBT and more pathology in the temporal and cingulate cortex compared to both other subtypes.

Compared to the other subtypes, individuals in the S3 (Brainstem-early/OBT-later) subtype seem to develop mild LB pathology across most brain regions early on, suggesting rapid spreading throughout the brain (Fig. 2). In contrast, both the S1 (OBT-early/Limbic-early) and S2 (OBT-early/Brainstem-early) seem to sequentially accumulate extensive pathology, affecting one region before another is affected. This is further illustrated by the finding that, in early disease stages (SuStaIn stage < 20), individuals assigned to the S3 (Brainstem-early/OBT-later) subtype have on average a larger number of brain regions that show non-zero LTS, and fewer regions that show severe or very severe LTS (Fig. 3B).

### Disease subtyping and staging

The three-subtype model was applied, classifying subjects into one of 40 stages along one of the three subtypes (Fig. 2). Out of 814 individuals, 13 (1.6%) were assigned a SuStaIn stage of 0, indicating they likely did not have sufficient LTS pathology to be subtyped and were excluded from subsequent analyses. LTS pathology in these cases is described in Table S3. Overall, cases showed a high certainty of subtype assignment (median = 97.7%, IQR = 73.1–100%), with lowest confidence at late stages where subtypes are more similar (Figure S4). For subsequent analyses, only cases with confident subtype assignment (>50% probability) were included (n = 781 [97.5%]). Of those, the majority were assigned to the S1 (OBT-early/Limbic-early) subtype (475 [60.8%]), followed by the S2 (OBT-early/Brainstem-early) (165 [21.1%]), and S3 (Brainstem-early/OBT-later) (141 [18.1%]) subtypes. Stage assignment was largely evenly distributed, with slightly more cases assigned to early stages and fewer to late stages (Fig. S5). Since the S1 (OBT-early/Limbic-early) and S2 (OBT-early/Brainstem-early) subtypes were identical

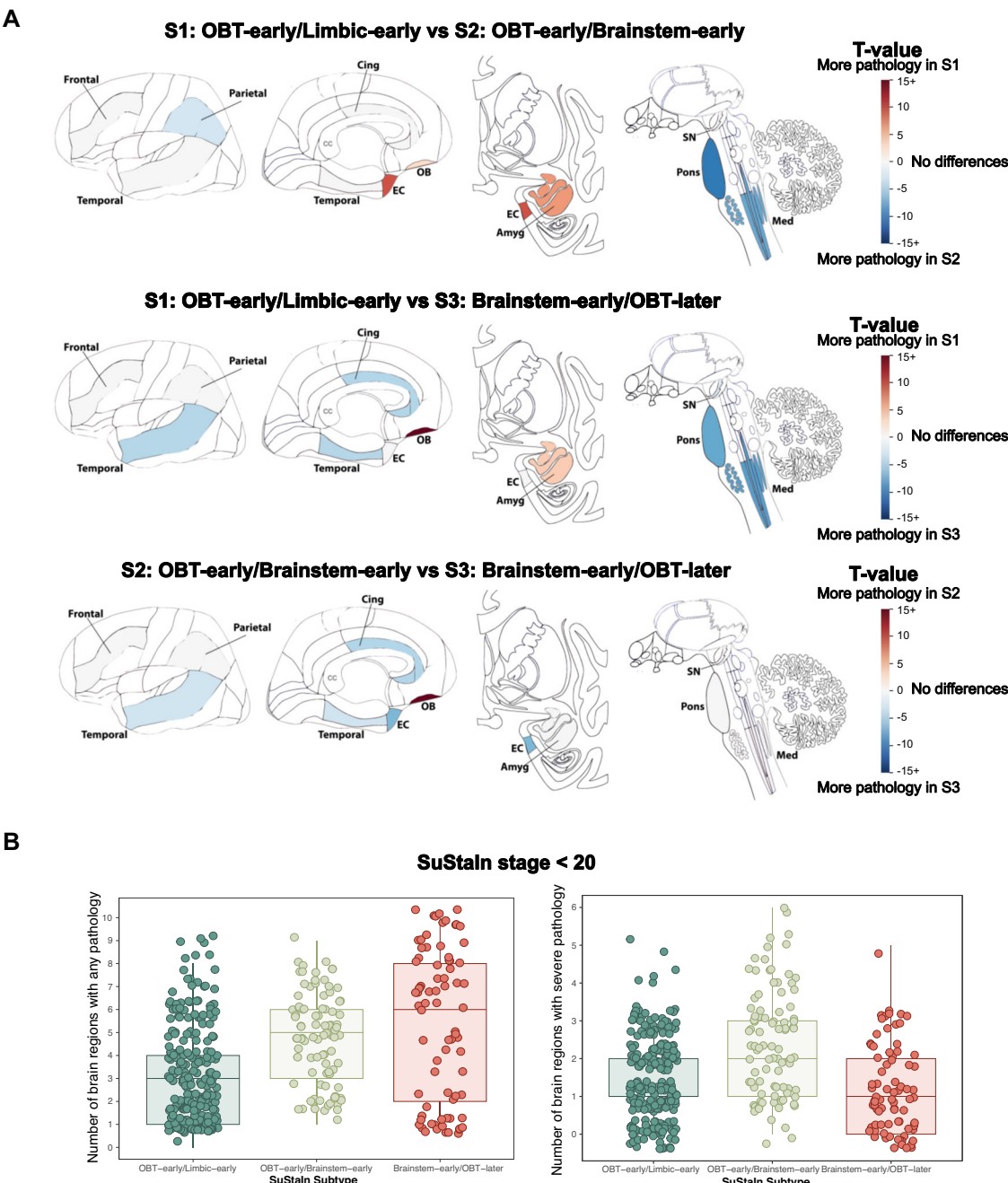

**Fig. 3 | Regional burden of LB pathology differs between subtypes. A** T-maps adjusted for SuStaIn stage and multiple comparisons, showing regions that are significantly different between LB subtypes. For visibility, *t*-values are shown on a scale from 15 to −15, but can represent values below and above as indicated by the 15+. Non-significant regions are shown in white. Brain schematics were generated using: https://github.com/AllenInstitute/hba_brain_schematic (Copyright © 2023. Allen Institute. All rights reserved.). **B** Number of brain regions affected by any or severe pathology in SuStaIn stages <20 (*n* = 420). The left panel indicates the number of brain regions displaying any Lewy body pathology (density score > 0) and the right panel indicates the number of brain regions with severe or very severe Lewy body pathology (density score > 2). Boxplots show the median, lower, and upper quartiles with whiskers representing minimum and maximum values. Amyg amygdala, Cing anterior cingulate, EC entorhinal cortex, Med medulla, OBT olfactory bulb and tract, SN substantia nigra, SuStaIn Subtype and Stage Inference.

in stage 1–3, SuStaIn assigned the more common subtype (i.e., S1) (see Methods).

**Younger age at death is associated with more advanced pathology across LB subtypes**

Figure 4 shows the univariate relationship between SuStaIn stage and age at death (*n* = 781) and total LB pathology (*n* = 673). Overall, the correlation between SuStaIn stage and total LB pathology was virtually collinear (S1: *r* = 0.99, *p* < 0.001; S2: *r* = 0.98, *p* < 0.001; S3: *r* = 0.98,

*p* < 0.001), reflecting as expected more advanced pathology with increasing SuStaIn stage. LB pathology was previously reported to be inversely correlated with age at death[18], and this association was presently observed across all three subtypes (S1: *r* = −0.14, *p* = 0.002; S2: *r* = −0.30, *p* < 0.001; S3: *r* = −0.18, *p* = 0.030). The interaction between SuStaIn subtype and stage on age at death was not significant (*p* = 0.07), although there was a trend towards a stronger relationship between stage and age for the S2 (OBT-early/Brainstem-early) subtype.

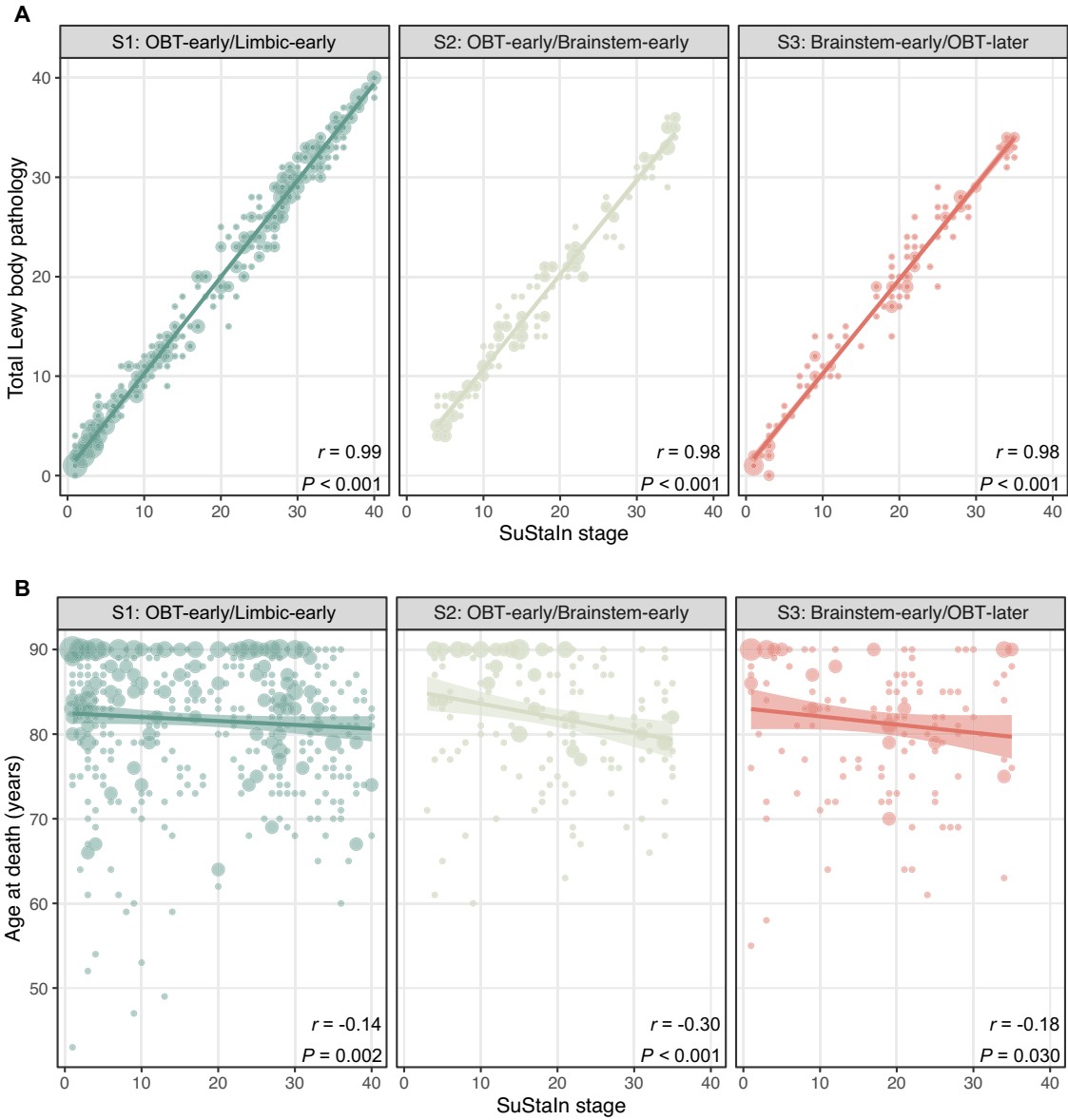

**Fig. 4 | SuStaIn stage is inversely correlated to age at death.** Two-sided spearman correlations between SustaIn stage and (**A**) total Lewy body pathology and (**B**) age at death. Total Lewy body pathology was defined by the sum of the α-synuclein density scores across 10 brain regions and was available for 673 (86.2%) cases. SuStaIn stage was strongly positively correlated to total Lewy body pathology and negatively associated with age in all subtypes. Error bands represent the standard error. Dot size represents the number of cases, with larger dots indicating larger sample size. OBT olfactory bulb and tract, SuStaIn Subtype and Stage Inference.

## Data-driven LB subtypes agree with a previous LB disease staging system

For 777 (99.5%) cases, classification according to the previously established Unified Staging System for Lewy Body Disorders (USSLB) was available (Fig. 5A)[10,18]. Briefly, Stage I includes cases with olfactory bulb pathology only, diverging into two pathways with Stage IIa reflecting brainstem-predominant involvement and Stage IIb limbic-predominant involvement. Stage IIa and IIb converge into Stage III, which includes cases with comparable levels of LB pathology in brainstem and limbic regions. Finally, the USSLB concludes with Stage IV, showing substantial involvement of at least one neocortical region. SuStaIn-inferred trajectories showed good agreement with the USSLB, whilst providing more comprehensive information. SuStaIn stage correlated to USSLB stage, with individuals in early SuStaIn stages (1–13) most often being classified as early USSLB stages (I, IIa, and IIb), and individuals in late SuStaIn stages (28–40) as late USSLB stages (III and IV) (Fig. S6).

Similar to the parallel brainstem- and limbic-predominant USSLB pathways, SuStaIn identified one limbic- and two brainstem-first subtypes. Individuals classified as USSLB stage IIa (brainstem-predominant) were most frequently assigned to the S2 (OBT-early/ Brainstem-early) (50%) and S3 (Brainstem-early/OBT-later) (41.3%) SuStaIn subtypes, while USSLB stage IIb (limbic-predominant) cases were most often assigned to the S1 (OBT-early/Limbic-early) SuStaIn subtype (89.8%) (Fig. 5B, C).

We confirm here, as first stated in the USSLB, that, in most subjects, LB pathology starts in the olfactory bulb. SuStaIn adds to this, however, an OBT-later subtype with earliest involvement of brainstem regions. None of the USSLB Stage 1 (olfactory bulb only) cases were assigned to the S3 (Brainstem-early/OBT-later) SuStaIn subtype (Fig. 5B, C). Instead, all USSLB stage 1 cases were assigned to the S1 (OBT-early/Limbic-early) subtype. In addition, SuStaIn provides a data-driven hypothesis as to where initial pathology started, even when pathology is widespread.

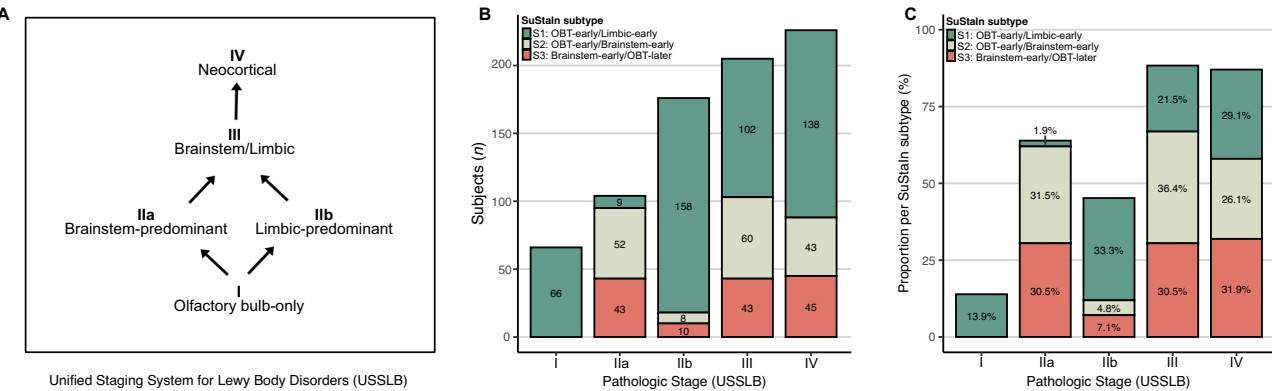

**Fig. 5 | Data-driven LB subtypes resemble the previous Unified Staging Scheme for Lewy Body Disorders.** Staging of subjects ($n = 777$) according to the Unified Staging Scheme for Lewy Body Disorders (**A**) across subtypes, shown as number of subjects (**B**) or proportion within each SuStaIn subtype (**C**).

## Early limbic deposition is associated with a clinicopathological diagnosis of AD

To characterize LB subtypes, we used separate logistic regressions adjusted for age, sex, and SuStaIn stage to compare them on clinicopathological diagnosis, assessed *postmortem* (Table 1). Compared to both other subtypes, the S1 (OBT-early/Limbic-early) subtype had a higher proportion of AD (vs S2: $\beta = 2.04$, $p < 0.001$; vs S3: $\beta = 3.41$, $p < 0.001$) and a lower proportion of LB clinicopathological diagnoses (Table 1; Fig. 6A). More specifically, the S1 (OBT-early/Limbic-early) subtype consisted of fewer PD ($\beta = -0.99$, $p < 0.001$), mixed AD/DLB ($\beta = -1.00$, $p = 0.003$), and other non-AD/LB ($\beta = -1.61$, $p = 0.002$) cases compared to the S3 (Brainstem-early/OBT-later) subtype, and fewer ILBD cases compared to both other subtypes (vs S2: $\beta = -2.18$, $p = 0.030$; vs S3: $\beta = -1.36$, $p < 0.001$). Among the two brainstem-predominant subtypes, the S2 (OBT-early/Brainstem-early) subtype had a higher proportion of AD ($\beta = 1.36$, $p = 0.001$) and ILBD ($\beta = 0.83$, $p = 0.030$) diagnosis and a lower proportion of PD ($\beta = -0.80$, $p = 0.018$) diagnosis.

## Extent of AD pathology distinguishes between clinicopathological Lewy body diagnoses

Next, we investigated whether the extent (i.e., stage) and/or the spatial distribution (i.e., subtype) of pathology could discriminate between clinicopathological PD ($n = 242$) and DLB ($n = 145$) diagnosis, also including cases with co-occurring AD. SuStaIn subtype, stage, and total LB pathology were not predictive of LB diagnosis. DLB diagnosis was associated with increased AD pathology burden (plaques: $\beta = 0.23$, $p < 0.001$; neurofibrillary pathology: $\beta = 0.31$, $p < 0.001$). In a combined model of SuStaIn subtype and total LB, amyloid plaques, and NFT score, only extent of pathology was able to discriminate between PD and DLB, with less LB pathology and more AD pathology in DLB cases (LB: $\beta = -0.05$, $p < 0.035$; plaques: $\beta = 0.19$, $p < 0.001$; neurofibrillary pathology: $\beta = 0.22$, $p < 0.001$).

## LB subtypes are characterized by distinct clinical, genetic, and neuropathological profiles

We compared subtypes on several demographic, pathological, and clinical variables of interest, assessed closest to time of death (Table 1; Fig. 6B-H). Age at death, sex, years of education, and *postmortem* interval did not differ between subtypes. The S1 (OBT-early/Limbic-early) subtype consisted of more *APOE* ε4 carriers (vs S2: $\beta = 0.40$, $p = 0.035$; vs S3: $\beta = 0.68$, $p = 0.001$; Fig. 6B) and had more plaque and neurofibrillary pathology than the other two subtypes (vs S2: $\beta_{plaques} = 2.20$, $p_{plaques} < 0.001$, $\beta_{neurofibrillary\ pathology} = 2.65$, $p_{neurofibrillary\ pathology} < 0.001$; vs S3: $\beta_{plaques} = 4.77$, $p_{plaques} < 0.001$, $\beta_{neurofibrillary\ pathology} = 4.12$, $p_{neurofibrillary\ pathology} < 0.001$; Fig. 6C, D). Individuals assigned to the S2 (OBT-early/Brainstem-early) subtype had a greater

total brain LB ($\beta = 0.47$, $p = 0.013$), plaque ($\beta = 2.57$, $p < 0.001$), and neurofibrillary ($\beta = 1.47$, $p = 0.003$) burden than the S3 (Brainstem-early/OBT-later) subtype. In addition, the S3 (Brainstem-early/OBT-later) subtype showed more severe substantia nigra depigmentation as compared to the other two subtypes (vs S1: SN level 2: $\beta = 3.10$, $p = 0.001$, SN level 3: $\beta = 4.35$; $p < 0.001$; vs S2: SN level 3: $\beta = 3.20$ $p = 0.003$; Fig. S7). No differences in GBA mutation status were observed (Fig. S7).

Regarding clinical symptoms, individuals in the S1 (OBT-early/Limbic-early) subtype performed worse on global cognition as compared to the S2 (OBT-early/Brainstem-early) ($\beta = -6.08$, $p < 0.001$) and S3 (Brainstem-early/OBT-later) ($\beta = -3.30$, $p < 0.001$) subtype (Fig. 6E). The S3 (Brainstem-early/OBT-later) subtype showed worse global cognition than the S2 (OBT-Brainstem-first) subtype ($\beta = -2.78$, $p = 0.019$) and worse motor symptoms (vs S1: $\beta = 5.56$, $p = 0.043$; vs S2: $\beta = 9.72$, $p = 0.003$; Fig. 6F) and sense of smell (vs S1: $\beta = -2.40$, $p = 0.047$; vs S2: $\beta = -3.54$, $p = 0.009$; Fig. 6H) compared to the other two. The S2 (OBT-early/Brainstem-early) subtype had a higher proportion of the Tremor Dominant phenotype of Parkinson's disease compared to the S3 (Brainstem-early/OBT-later) subtype ($\beta = 0.87$, $p = 0.040$) (Fig. 6G).

When additionally adjusting for total cortical plaque load, the LB subtype was no longer significantly associated with *APOE* ε4 carriership. Likewise, differences in tau pathology and global cognition were no longer significant for the S2 (OBT-early/Brainstem-early) and S3 (Brainstem-early/OBT-later), and S1 (OBT-early/Limbic-early) and S3 (Brainstem-early/OBT-later) subtypes, respectively.

A sliding window approach revealed that some of these subtype differences were dynamic across SuStaIn stages, likely driven by the higher prevalence of AD relative to LB diseases in early stages and vice versa in late stages (Fig. S8). This trend was seen across S1 (OBT-early/Limbic-early) and S2 (OBT-early/Brainstem-early) but not so much S3 (Brainstem-early/OBT-later).

## LB subtypes show different rates of clinical progression

Longitudinal clinical data with at least two timepoints was available in a subset of cases for the MMSE ($n = 211$, 2–16 assessments across $3.0 \pm 3.0$ years), UPDRS-III ($n = 303$, 2–18 assessments across $4.3 \pm 4.2$ years), and UPSIT ($n = 101$, 2-5 assessments across $3.7 \pm 3.8$ years). For the MMSE and UPDRS-III, linear models with linear and quadratic terms fitted the data better compared to linear models with only linear terms (MMSE: $\Delta AIC = -120.9$, $p < .001$; UPDRS-III: $\Delta AIC = -56$, $p < .001$). While the quadratic terms were not significant, subjects in the S2 (OBT-early/Brainstem-early) subtype showed attenuated decline on global cognition and motor symptoms as compared to the other subtypes (MMSE: vs S1: $\beta = 0.99$, $p < 0.001$; vs S3: $\beta = 0.85$, $p = 0.026$; Fig. 6I; UPDRS-III: vs S1: $\beta = -1.26$, $p = 0.006$; vs S3: $\beta = -1.75$, $p = 0.009$; Fig. 6J). No

**Table 1 | Participant characteristics across inferred Lewy body subtypes**

| | SuStain Subtype | | | Subtype comparisons | | | | | |
|---|---|---|---|---|---|---|---|---|---|
| | S1: OBT-early/Limbic-early | S2: OBT-early/Brainstem-early | S3: Brainstem-early/OBT-later | S1 vs S2 | | S1 vs S3 | | S2 vs S3 | |
| | | | | β (se) | P | β (se) | P | β (se) | P |
| N (%) | 475 | 165 | 141 | - | - | - | - | - | - |
| Age at death (years) | 81.7 (8.0) | 82.2 (7.1) | 81.4 (7.8) | −0.58 (0.70) | 0.401 | 0.32 (0.74) | 0.664 | 0.91 (0.89) | 0.306 |
| Female (%) | 186 (39) | 62 (37.6) | 58 (41.1) | 0.07 (0.19) | 0.702 | −0.08 (0.20) | 0.693 | −0.15 (0.24) | 0.528 |
| Education, years[a] | 15.0 (2.9) | 14.7 (2.8) | 14.9 (2.6) | 0.36 (0.27) | 0.188 | 0.02 (0.31) | 0.945 | −0.34 (0.36) | 0.349 |
| APOE ε4 carriers (%)[b] | 217 (45.7) | 59 (35.8) | 43 (30.5) | 0.40 (0.19) | 0.035* | 0.68 (0.21) | 0.001* | 0.28 (0.25) | 0.252 |
| Clinicopathological diagnosis | | | | | | | | | |
| AD, n (%) | 239 (50.3) | 29 (17.6) | 12 (8.5) | 2.04 (0.27) | <0.001* | 3.41 (0.37) | <0.001* | 1.36 (3.41) | 0.001* |
| PD, n (%) | 89 (18.7) | 33 (20.0) | 46 (32.6) | −0.19 (0.24) | 1.000 | −0.99 (0.24) | <0.001* | −0.80 (0.28) | 0.018* |
| DLB, n (%) | 6 (1.3) | 5 (3.0) | 7 (5.0) | −0.94 (0.62) | 0.448 | −1.50 (0.57) | 0.063 | −0.56 (0.60) | 0.812 |
| Mixed AD and PD, n (%) | 38 (8.0) | 24 (14.5) | 12 (8.5) | −1.06 (0.31) | 0.070 | −0.45 (0.37) | 0.539 | 0.61 (0.40) | 0.445 |
| Mixed AD and DLB, n (%) | 67 (14.1) | 29 (17.6) | 31 (22) | −0.53 (0.27) | 0.185 | −1.00 (0.28) | 0.003* | −0.46 (0.32) | 0.355 |
| ILBD, n (%) | 25 (5.3) | 38 (23.0) | 21 (14.9) | −2.18 (0.32) | 0.042* | −1.36 (0.35) | 0.001* | 0.83 (0.35) | 0.042* |
| Other, n (%) | 11 (2.3) | 7 (4.2) | 12 (8.5) | −1.16 (0.53) | 0.098 | −1.61 (0.46) | 0.003* | −0.45 (0.52) | 0.905 |
| Total Lewy body density[c] | 18.1 (12.2) | 18.0 (9.3) | 17.2 (10.3) | −0.28 (0.15) | 0.059 | 0.19 (0.16) | 0.226 | 0.47 (0.19) | 0.013* |
| Total plaque load[d] | 10.7 (5.2) | 8.5 (5.6) | 6.0 (5.8) | 2.20 (0.49) | <0.001* | 4.77 (0.52) | <0.001* | 2.57 (0.62) | <0.001* |
| Total neurofibrillary tangle load[e] | 10.1 (4.5) | 7.4 (4.4) | 6.1 (4.2) | 2.65 (0.38) | <0.001* | 4.12 (0.40) | <0.001* | 1.47 (0.49) | 0.003* |
| Postmortem interval (hours) | 5.8 (11.2) | 5.5 (8.5) | 4.3 (5.3) | 0.30 (0.88) | 0.737 | 1.68 (0.94) | 0.074 | 1.38 (1.12) | 0.218 |
| MMSE score[f] | 15.2 (9.5) | 21.4 (8.5) | 18.5 (9.1) | −6.08 (0.89) | <0.001* | −3.30 (0.99) | <0.001* | 2.78 (1.18) | 0.019* |
| Δt MMSE and death (months) | 21.7 (2.2) | 21.2 (21.7) | 21.6 (22.1) | - | - | - | - | - | - |
| Motor UPDRS score[g] | 28.8 (22.4) | 20.9 (21.6) | 32.3 (22.7) | 4.16 (2.59) | 0.109 | −5.56 (2.74) | 0.043* | −9.72 (3.24) | 0.003* |
| Δt UPDRS and death (months) | 17.4 (17.2) | 20.0 (25.2) | 15.5 (18.6) | - | - | - | - | - | - |
| UPSIT score[h] | 16.4 (7.1) | 19.5 (8.4) | 15.9 (8.2) | −1.13 (0.98) | 0.248 | 2.40 (1.20) | 0.047* | 3.54 (1.34) | 0.009* |
| Δt UPSIT and death (months) | 3.5 (2.4) | 2.9 (2.0) | 3.3 (2.8) | - | - | - | - | - | - |

Descriptive characteristics of the S1 (OBT-early/Limbic-early), S2 (OBT-early/Brainstem-early), and S3 (Brainstem-early/OBT-later) subtypes. Data are shown as mean (standard deviation) unless otherwise specified. Output of linear (continuous variables) and logistic (categorical variables) regression models comparing SuStain subtypes are shown as estimates (standard error) and P value. *Indicates significant differences (P < 0.05). Reported P values of models predicting clinicopathological diagnosis are FDR corrected for n = 7 comparisons.

MMSE Minimal Mental State Examination, OBT olfactory bulb and tract, PMI Postmortem interval, SuStain subtype and stage inference, UPDRS Unified Parkinson's Disease Rating Scale.

[a] n S1 (OBT-early/Limbic-early) = 389; n S2 (OBT-early/Brainstem-early) = 139; n S3 (Brainstem-early/OBT-later) = 101.
[b] n S1 (OBT-early/Limbic-early) = 471; n S2 (OBT-early/Brainstem-early) = 163; n S3 (Brainstem-early/OBT-later) = 140.
[c] Sum of the regional density scores (range 0–40). n S1 (OBT-early/Limbic-early) = 414; n S2 (OBT-early/Brainstem-early) = 139; n S3 (Brainstem-early/OBT-later) = 120.
[d] n S1 (OBT-early/Limbic-early) = 471; n S2 (OBT-early/Brainstem-early) = 162; n S3 (Brainstem-early/OBT-later) = 139.
[e] n S1 (OBT-early/Limbic-early) = 469; n S2 (OBT-early/Brainstem-early) = 160; n S3 (Brainstem-early/OBT-later) = 140.
[f] n S1 (OBT-early/Limbic-early) = 407; n S2 (OBT-early/Brainstem-early) = 136; n S3 (Brainstem-early/OBT-later) = 103.
[g] n S1 (OBT-early/Limbic-early) = 230; n S2 (OBT-early/Brainstem-early) = 83; n S3 (Brainstem-early/OBT-later) = 72.
[h] n S1 (OBT-early/Limbic-early) = 158; n S2 (OBT-early/Brainstem-early) = 62; n S3 (Brainstem-early/OBT-later) = 35.

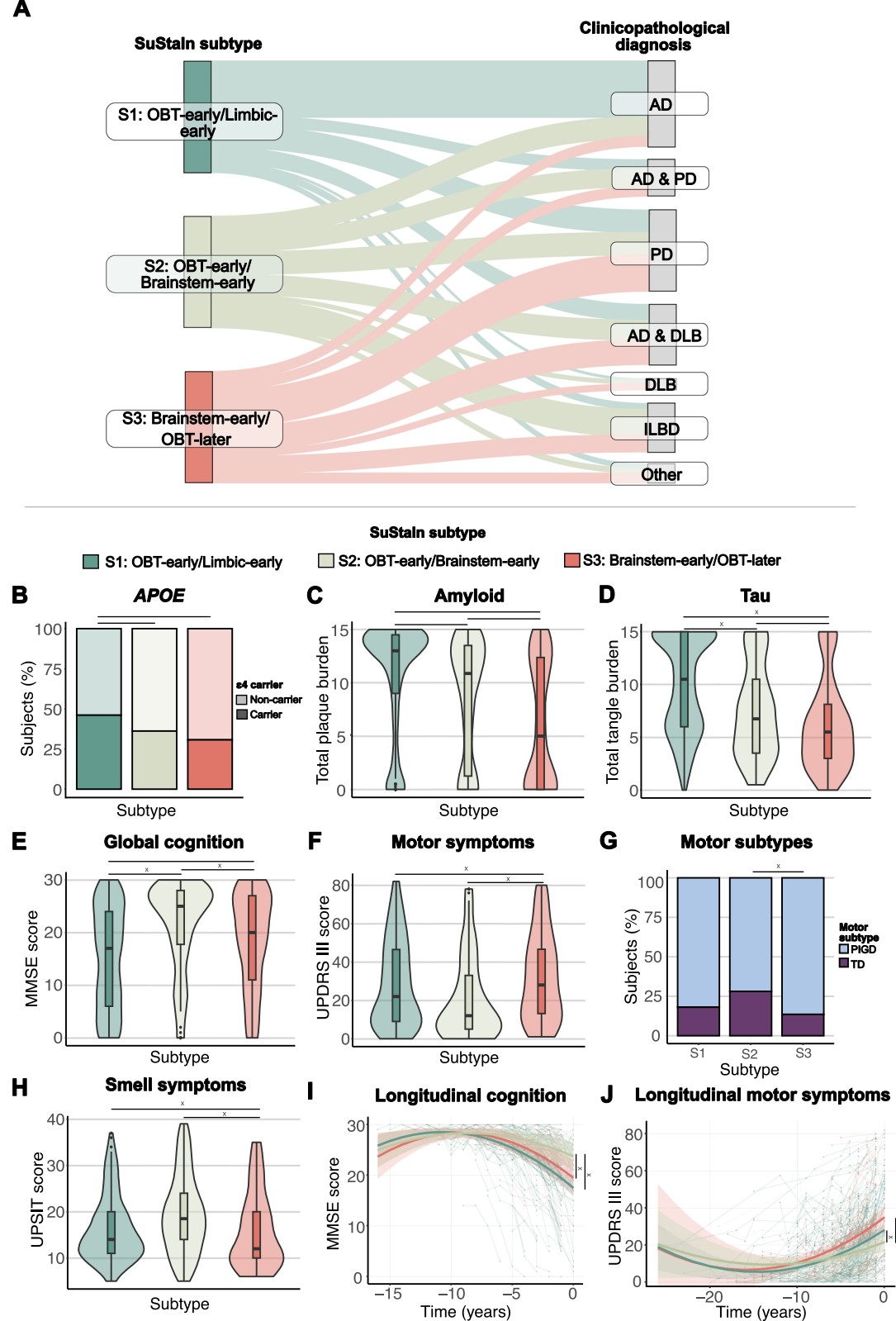

differences in longitudinal trajectories of sense of smell were observed. Adjusting for plaque burden did not change the results.

**Peripheral Lewy body pathology emerges earlier in brainstem-early subtypes**

For 236 subjects (S1: *n* = 165, S2: *n* = 52, S3: *n* = 19), a total score of non-brain LB pathology was computed by summing the *postmortem*

density scores of the spinal cord and peripheral regions, including the cervical, thoracic, lumbar, and sacral spinal cord gray matter, vagus nerve, submandibular gland, and esophagus, ranging from 0-28. Modeling total non-brain LB scores across SuStaIn stages revealed that while both brainstem-first subtypes showed non-brain LB burden from the earliest disease stages, the S1 (OBT-early/Limbic-early) subtype did not accumulate pathology until stages 10–15

**Fig. 6 | LB subtypes are characterized by distinct clinicopathological, genetic, and clinical characteristics.** Results of SuStaIn subtype comparisons. **A** Clinicopathological diagnosis in relation to SuStaIn subtype. The Sankey diagram shows the proportion (%) of SuStaIn subtypes across clinicopathological diagnoses. Subtype differences assessed with two-sided linear and logistic regression models, for continuous and categorical variables respectively, are shown for (**B**) *APOE* ε4 carriership (n = 774); **C** total *postmortem* plaque burden (n = 772); **D** *postmortem* neurofibrillary burden (n = 769); **E** MMSE measuring global cognition (n = 646); **F** UPDRS part III measuring motor symptoms (assessed off medication) (n = 385); **G** motor subtypes (n = 385); and (**H**) UPSIT measuring smell ability (n = 255). Clinical variables were measured closest to time of death. Horizontal lines reflect significant differences. Longitudinal trajectories of (**I**) MMSE and (**J**) UPDRS

part III in the different subtypes. Model-predicted associations are plotted for each subtype from linear mixed-effect models including a polynomial (non-linear) term for time. Covariates were age, sex, education and SuStaIn stage. Time 0 indicates baseline anchored to the date of autopsy. Individual trajectories are shown in the background. Vertical lines represent significant differences (all p < 0.05). x indicates differences that persist after adjusting for plaque burden. Boxplots show the median, lower, and upper quartiles with whiskers representing minimum and maximum values. AD Alzheimer's disease, DLB dementia with Lewy bodies, ILBD incidental Lewy body disease, MMSE Mini-Mental State Examination, OBT olfactory bulb and tract, PD Parkinson's disease, PIGD Postural Instability and Gait difficulties, SuStaIn Subtype and Stage Inference, TD tremor dominant, UPDRS Unified Parkinson's Disease Rating Scale, UPSIT Smell Identification Test.

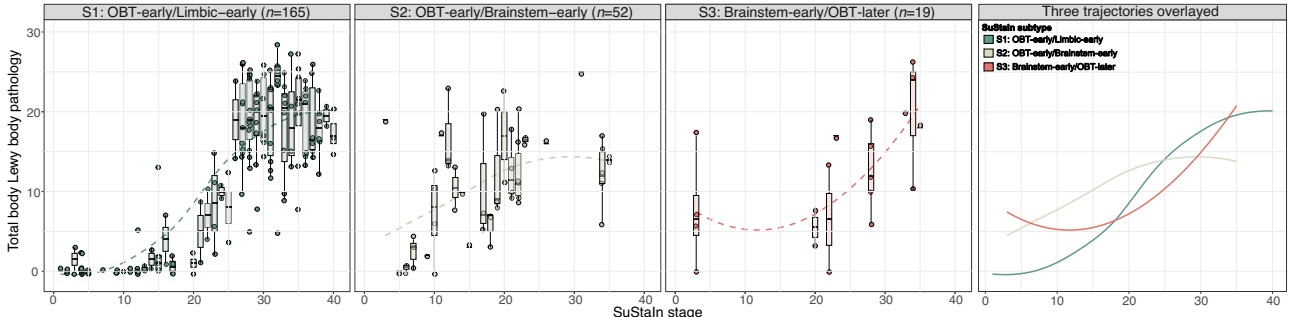

**Fig. 7 | Early LB pathology in brainstem regions is associated with substantial non-brain LB pathology in early SuStaIn stages.** Total peripheral Lewy body pathology across SuStaIn stages, shown by SuStaIn subtype. Total pathology scores were computed as the sum of the α-synuclein density scores (0–4) assessed in the cervical, thoracic, lumbar, and sacral spinal cord gray matter, vagus nerve, submandibular gland, and esophagus, with higher scores indicating more severe

pathology. Dashed lines represent modeled trajectories computed with LOESS regressions. The right-most panel shows the 4 inferred trajectories overlaid. Boxplots show the median, lower, and upper quartiles with whiskers representing minimum and maximum values. OBT olfactory bulb and tract, SuStaIn Subtype and Stage Inference.

(Fig. 7). Even though data was scarce and variable between subjects, visually comparing the brainstem subtypes showed that the S3 (Brainstem-early/OBT-later) subtype might have a tendency to start out with more pathology in early stages (<10) compared to the S2 (OBT-early/Brainstem-early) subtype (Figure S9). In both the S1 (OBT-early/Limbic-early) and S2 (OBT-early/Brainstem-early) subtypes, non-brain LB pathology was found to reach a plateau around stage 25 and stage 20, respectively.

## Discussion

In this study, applying data-driven disease progression modeling to a large *postmortem* dataset of 814 subjects supported the existence of multiple distinct spatiotemporal subtypes of Lewy body progression, showing good agreement with previous staging systems. Our analysis suggests that most individuals (82%) showed the earliest pathology in the olfactory bulb, followed by accumulation in either limbic or brainstem regions, while the remaining subjects exhibited heavier early abnormalities primarily in brainstem regions. Early LTS deposition in limbic regions was associated with AD-like characteristics, including a greater proportion of *APOE* ε4 carriers, more AD pathology, and worse cognitive functioning compared to the other two subtypes. Similarly, initial pathology in olfactory bulb and brainstem regions was associated with attenuated clinical decline over time. Finally, supporting prior observations[26], a comparison of non-brain LB pathology across subtypes showed that individuals with early pathology in brainstem regions exhibited substantial LB pathology in the spinal cord and peripheral regions from the earliest disease stages, with a tendency towards more non-brain pathology in those subjects where the brainstem is the first region to be affected, although it should be noted that few datapoints were available in this subtype. In contrast, the limbic-early subtype seemed to start accumulation of non-brain LBs later in the disease trajectory.

Previous studies have reported a high prevalence of concomitant LB and AD pathology, with coexisting AD pathology in up to 50% of PD cases with dementia and more than 75% of DLB cases, and LTS co-pathology in ~60% of AD cases[24,27–36]. LB co-pathology in AD is most frequently observed in the amygdala and can also occur in neocortical and brainstem areas, although to a lesser extent[37–40]. In line with this notion, SuStaIn identified a limbic-predominant subtype, where earliest abnormalities were observed in the olfactory bulb and amygdala, while more severe depositions in brainstem and neocortical regions were observed in later stages only. Importantly, individuals assigned to the limbic-predominant subtype were more frequently diagnosed *postmortem* with AD as compared to the other two subtypes. In addition, early limbic pathology was associated with a higher proportion of *APOE* ε4 carriers, increased plaque and neurofibrillary load, and worse performance on global cognition during life. Taken together, our results suggest that the S1 (OBT-early/Limbic-early) subtype might primarily reflect cases with co-occurring AD and LB pathology. A potential underlying mechanism is that plaques may have a seeding effect on α-synuclein as described previously[28,41], possibly facilitating LB pathology with a trajectory of spread as seen in the limbic-early SuStaIn subtype.

In contrast, PD and DLB cases were more frequently assigned to the two subtypes with early brainstem pathology. Across these two brainstem-first subtypes, the S3 (Brainstem-early/OBT-later) subtype had a higher proportion of clinicopathological PD and a lower proportion of ILBD, implying that these subjects tend to have a more advanced clinical presentation. Additionally, we showed that, rather than the spatial distribution, the extent of pathology was able to distinguish between clinicopathological DLB and PD. This was most profoundly observed for plaques and neurofibrillary inclusions, which is in line with studies showing a greater AD pathological burden in DLB compared to PD cases[42–46]. Seeing the overlap in clinical and

neuropathological findings between LB diseases, it has previously been suggested that PD and DLB can be considered subtypes within an α-synuclein-associated disease spectrum, rather than separate entitites[47]. Our results partially corroborate this framework by demonstrating a relative lack of evidence for distinct spatiotemporal PD and DLB pathological subtypes, with heavier AD pathological burden in DLB but not the spatiotemporal patterns of LB pathology distinguishing the groups.

Similar to the USSLB, we observed that in the majority of cases, pathology starts in the olfactory bulb. The selective vulnerability of the olfactory bulb to LTS has been thoroughly discussed previously and is hypothesized to be attributable to cellular characteristics and proximity to the spatial/external environment[15,18,48]. While underlying mechanisms of selective cell type vulnerability remain largely unknown, it has been speculated that morphological properties, such as long, poorly myelinated axons that are often affected by LBs, might have an important role[48,49]. Interestingly, SuStaIn identified a potential third subtype that had early brainstem involvement, relative to the olfactory bulb. This subtype was associated with (i) non-brain LB pathology in early SuStaIn stages, (ii) more nigral neuron loss, (iii) worse motor and smell impairment, (iv) a higher proportion of the PIGD motor-phenotype, and (v) similar rates of clinical progression on cognitive and motor measures as the AD-associated S1 (OBT-early/Limbic-early) subtype, despite having the lowest and highest proportion of AD and PD subjects, respectively. The finding of worse smell symptoms, higher proportion of PIGD phenotype and faster clinical decline is in line with symptoms expected in a hypothetical body-first LB type (Fig. 1C)[19,20,50–53], as described in previous literature[51]. However, the body-first subtype has been debated, as a detailed comparison of vagus nerve LB pathology with brain LB pathology failed to find any cases where LB pathology in the vagus nerve or stomach was present in the absence of brain LB pathology. In addition, the AZSAND/BBDP has never observed a single case with LB pathology present anywhere in the body but not in the brain[17,26]. While our data cannot lend any definitive evidence for or against a body-first subtype, the data clearly highlight that (i) a proportion of the population accumulates early and substantial LB in peripheral regions, while another proportion does not; (ii) that these populations are associated with distinct CNS patterns of LB pathology; and (iii) that the populations appear to present with distinct clinical presentations.

Interestingly, despite relatively late involvement of olfactory regions, the S3 (Brainstem-early/OBT-later) subtype exhibited worse smell impairment compared to both other subtypes. Of note, previous studies have shown that it is not the extent of olfactory bulb LB pathology, but rather the total brain LB pathology, that is associated with smell function[54,55], for which subtype comparisons in this study were adjusted. A possible explanation for worse smell impairment in the S3 subtype that has been proposed before in light of the brain- vs body-first hypothesis, is that patients with the hypothesized brain-first LB type might show initial asymmetrical olfactory pathology, resulting in initially undetectable smell dysfunction, since the unaffected nostril might be able to compensate for this smell deficit[51,56]. Alternatively, it is possible that the lower smell test performance is a result of the higher proportion of clinicopathologically defined PD cases in the S3 subtype, with several studies estimating the presence of smell impairments in more than 90% of patients with PD[57,58]. This would also potentially explain the worse motor symptoms observed in this subtype.

SuStaIn stage was negatively correlated to age in all three LB subtypes, which has been previously observed in LB diseases[18] and other proteinopathies, such as tau in AD[59–62] and TDP-43 in the frontotemporal lobar degeneration and the amyotrophic lateral sclerosis spectrum[23]. The younger age at death of individuals who are at more advanced disease stages could be due to a more aggressive (often early onset) disease course, consequently dying at a younger age. In addition, patients with an early onset are more likely to die as a consequence of the disease and therefore acquire a greater neuropathological burden.

This theory is supported by observations in AD, with early-onset patients displaying more plaques and tangles at autopsy, despite having a younger age at death[60,63,64]. This has also been strongly supported by in vivo neuroimaging studies, showing higher tau positron emission tomography (PET) burden and accumulation in early-onset as compared to late-onset AD[59,62,65–67]. Alternatively, patients with extensive LB pathology at a younger age might have a more pure LB disease, whereas those that are older might have died from other (non-neurodegenerative) diseases with lower levels of LB co-pathology or a LB disorder with other co-pathologies, leading to a greater cumulative pathological burden. Finally, younger age has been previously found to be associated with a higher cognitive reserve[68], implying that younger subjects might be able to cope with a greater burden of pathology.

Strengths of this study include the use of a probabilistic data-driven approach to model disease trajectories from cross-sectional data and the large dataset of well-characterized *postmortem* cases. Several limitations also have to be addressed. First, data availability regarding spinal cord and peripheral LB measures was limited, warranting careful interpretation. Second, non-brain regions were not as frequently stained for LTS in cases where brain LB pathology was not present, potentially introducing bias. Third, as the cases were autopsied between 1997 and 2021, AD and DLB clinicopathological diagnoses are based on older iterations of neuropathological criteria, although changes in criteria are minor[4,25,69]. Fourth, we found that the SuStaIn subtypes corresponded closely with those originally developed in the USSLB, which was based on the same cohort as was used in the current study, hence potentially exhibiting circularity. However, one could also argue that two independent approaches leading to comparable outcomes supports the validity of our findings. In addition, the USSLB has been found, in a multicenter study, to be superior to the Braak system but similar to other more recently devised systems datasets at classifying cases with Lewy body pathology to a disease stage[70,71]. Finally, the cross-sectional study design warrants future validation in longitudinal in vivo datasets, once α-synuclein PET imaging is available[72].

To conclude, by applying a data-driven modeling approach to *postmortem* density scores assessed in a large number of regions, we show significant heterogeneity in LB spreading trajectories, supporting and extending on previous literature. Specifically, in line with previous studies, we identified two subtypes that show earliest pathology in the olfactory bulb followed by either limbic or brainstem regions. Furthermore, we describe an additional subtype with initial quantitatively greater involvement of the brainstem, reaching severe levels of olfactory bulb pathology relatively later. These different disease progression patterns were associated with distinct demographic, genetic, pathological, and clinical characteristics. Most notably, the subtype with early limbic deposition seemed to be associated with an AD phenotype, implying that plaque accumulation may facilitate LB pathology in a pattern as seen in this SuStaIn subtype. Finally, early pathology in the brainstem was more likely to be associated with substantial LB pathology in the peripheral nervous system. Understanding disease progression is essential for providing insights into the pathogenesis of a disease and can, together with the development of in vivo α-synuclein PET tracers, potentially be used to support patient stratification in clinical trials.

## Methods
### Subjects and pathological assessments
814 neuropathological samples were selected from the Arizona Study of Aging and Neurodegenerative Disorders/Brain and Body Donation Program, part of the Banner Sun Health Research Institute (SHRI), which was approved by the SHRI Institutional Review Board[24]. All enrolled subjects signed an Institutional Review Board-approved informed consent. Procedures of brain harvesting, tissue preparation, staining, and diagnostic assessment have been described in detail previously[18,24]. Briefly, immunohistochemical α-synuclein stainings were performed with a polyclonal antibody raised against an α-synuclein

peptide fragment phosphorylated at serine 129 (pS129)[73,74]. For each autopsy case, 10 standard regions were sampled (excluding only damaged or missing regions) comprising the OBT, anterior medulla, anterior and mid-pons, substantia nigra, mid-amygdala, transentorhinal area, anterior cingulate gyrus, middle temporal gyrus, middle frontal gyrus, and inferior parietal lobule. Each region was graded for density of Lewy-type α-synuclein (LTS) by a single observer, according to a semi-quantitative rating scale ranging from 0 to 4, where 0 = none, 1 = mild, 2 = moderate, 3 = severe, and 4 = very severe pathology[25,71]. Example images of various density scores are shown for the OBT in Fig. S10. Since cases without any α-synuclein pathology in the brain do not add information to the SuStaIn model, only individuals with at least one region with a density score ≥1, were included in this study. In addition, individuals with a significant proportion of missing data (>3 [30%] ROIs) were excluded. To ensure that missingness did not influence the inferred pathological trajectories, we additionally applied the SuStaIn model to a subsample with complete data (N = 701).

For subsequent statistical analyses, most included subjects had available measures of AD pathology, including total plaque load (n = 771) and total neurofibrillary pathology (n = 769), measured as the sum of the semi-quantitative Consortium to Establish a Registry for Alzheimer's Disease (CERAD) density scores (0 = none, 1 = sparse, 2 = moderate, 3 = frequent) in standard regions of the frontal, temporal, and parietal lobes, hippocampal CA1 region, and entorhinal/transentorhinal region[75]. For 338 cases, peripheral LTS density data was utilized from up to 7 additional non-brain regions, including the cervical, thoracic, lumbar, and sacral spinal cord gray matter, vagus nerve, submandibular gland, and esophagus[17,24]. For 236 subjects with at least 5 peripheral measures, we used the Multivariate Imputation by Chained Equations (MICE) R package to impute missing scores with predictive mean matching single imputation[76]. Total non-brain LB pathology was measured as the sum of the 7 non-brain density scores.

A final clinicopathological diagnosis was assigned after death according to specific diagnostic criteria[24,25,77–83], incorporating research clinical data, the most recent medical records, and neuropathological examination. Cases with evidence of LB pathology at neuropathological examination but who did not meet clinicopathological diagnostic criteria for PD, DLB, or any other neurodegenerative disease, and who had neither parkinsonism or dementia, were labeled as having incidental LB disease (ILDB). Cases without a major neuropathological diagnosis and who had no LTS and neither parkinsonism or dementia, were classified as controls.

## Clinical assessments

Most cases underwent annual neuropsychological, neurological, and movement examinations. In the current study, we focused on measures of global cognition (Mini-Mental State Examination [MMSE]), neuromotor symptoms (assessed off medication with the Unified Parkinson's Disease Rating Scale [UPDRS] part III), and smell symptoms (University of Pennsylvania Smell Identification Test [UPSIT]). For cross-sectional analyses, scores closest to time of death were used ($n_{MMSE}$ = 646; $n_{UPDRS-III}$ = 385; $n_{UPSIT}$ = 255). Longitudinal data with at least two timepoints was available in a subset of cases for the MMSE (n = 211, 2–16 assessments across 3.0 ± 3.0 years), UPDRS-III (n = 303, 2–18 assessments across 4.3 ± 4.2 years), and UPSIT (n = 101, 2–5 assessments across 3.7 ± 3.8 years).

In addition, cases were classified as a motor or "Postural Instability and Gait difficulties" (PIGD) and "Tremor-Dominant" (TD) subtype based on items of the UPDRS assessed closest to death as previously described[84].

## Disease progression modeling using SuStaIn

Disease progression modeling was performed using the Ordinal SuStaIn implementation in PySuStaIn[21,22]. SuStaIn is a probabilistic machine learning algorithm that combines clustering and disease staging to characterize distinct spatial-temporal disease progression patterns from cross-sectional data. The ordinal implementation of SuStaIn is specifically designed for application to variables with ordered categories, such as neuropathological density scores and requires as input region-specific probabilities that a given density score is observed. We transformed density scores of 0, 1, 2, 3, and 4 into probabilities by estimating a normal distribution with a standard deviation of 0.5 around each score and normalizing by the sum of the probabilities (Table S4)[23]. Missing density scores were modeled as having an equal probability for each score (i.e., 20%). By setting equal probabilities, missing scores will not influence model fit, which is preferred over removing the subject altogether and losing information from the other values.

Ordinal SuStaIn was used to estimate multiple progression patterns, with a prespecified number of subtypes (i.e., up to 5 subtypes). The model was fit using 10,000 Markov chain Monte Carlo iterations. Subsequently, 10-fold cross-validation was used to estimate the cross-validation information criterion (CVIC) and log-likelihood across folds to assess the optimal number of patterns/subtypes in the data, defined as the lowest CVIC and highest log-likelihood[85]. Each case was assigned to a subtype (i.e., a progression pattern) and stage (i.e., proxy for progression along the inferred pathological trajectory). In case of similar progression patterns between subtypes, for instance in late stages where pathology is widespread, SuStaIn assigns the most common subtype based on the assumption that the dominant subtype is more probable. Ordinal SuStaIn models disease trajectories as the ordering in which different brain regions reach different density scores, hence in our study 40 stages (10 regions * 4 score transitions) were modeled.

## Statistical analyses

SuStaIn subtypes were compared on regional LB density scores with linear regression models adjusted for SuStaIn stage. Only subjects assigned to SuStaIn stage ≥1 and with a confident subtype assignment (>50%) were included in the following statistical analyses (n = 781). To assess univariate relationships between SuStaIn stage and age at death and total LB pathology (sum of the density scores in 10 brain regions), Spearman correlations were run. Of note, participants >90 years of age were classified as 90, to safeguard potential subject-identification. Separate linear, logistic, and multinomial regression models for cross-sectional continuous, categorical, and ordinal variables, respectively, were used to compare SuStaIn subtypes on (1) clinicopathological diagnosis of AD, PD, DLB, mixed AD-PD, mixed AD-DLB, ILBD, and a group of any other diagnosis; (2) demographics (age at death, sex, years of education, APOE ε4 carriership, GBA (glucocerebrosidase) mutation status, and postmortem interval); (3) postmortem pathology (total LB, plaque, and neurofibrillary burden, and substantia nigra (SN) depigmentation); and (4) clinical variables assessed closest to time of death (MMSE, UPDRS part III, UPSIT, and PIGD/TD phenotypes). All models were adjusted for age, sex, and SuStaIn stage and models regarding clinical variables were additionally adjusted for education and time interval to death. To investigate whether observed subtype differences were driven by amyloid plaque pathology, regression analyses were repeated while additionally controlling for total amyloid plaque burden. Using a sliding window approach, we examined whether subtype differences on clinicopathological diagnosis, MMSE, sex, APOE-ε4 carriership, and plaque and neurofibrillary burden varied by disease stage. More specifically, regression models comparing subtypes as described above were repeated across a window width of 10 SuStaIn stages, moving from 1 (early) to 40 (late), with a slide of 1 stage (i.e., 1–10, 2–11 … 30–40), resulting in 31 models. In a subset, subtype differences in longitudinal MMSE, UPDRS part III, and UPSIT change were examined using linear mixed models with and without a polynomial term for time. Model performance was compared using ANOVAs. Mixed models included random intercept and slope, the interaction between subtype and time (years) was used as predictor, with time 0 aligned with date of

autopsy, and covariates were age, sex, education, and SuStaIn stage. The predictive value of SuStaIn subtype and stage, total LB pathology, total plaque burden, and total neurofibrillary load in distinguishing between PD and DLB clinical diagnosis was assessed using separate age and sex adjusted logistic regressions and a combined model of total LB, plaque, and neurofibrillary pathology. Finally, locally estimated scatterplot smoothing (LOESS) regressions were used to assess total non-brain LB pathology across SuStaIn stages.

Analyses were performed in R version 4.2.0[86]. Significance was set at two-sided $P < 0.05$. $P$ values regarding regional LB density and diagnosis were false discovery rate corrected for multiple comparisons based on the number of models.

### Reporting summary

Further information on research design is available in the Nature Portfolio Reporting Summary linked to this article.

## Data availability

Anonymized data will be shared by request as long as data transfer is in agreement with USA legislation (Privacy Rule of the Health Insurance Portability and Accountability Act). Source data are provided with this paper.

## Code availability

Source code for the Ordinal SuStaIn algorithm is available at https://github.com/ucl-pond/pySuStaIn.

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

## Acknowledgements

Work at Lund University was supported by the Alzheimer's Association (SG-23-1061717), Swedish Research Council (2022-00775), ERA PerMed (ERAPERMED2021-184), the Knut and Alice Wallenberg foundation (2017-0383), the Strategic Research Area MultiPark (Multidisciplinary Research in Parkinson's disease) at Lund University, the Swedish Alzheimer Foundation (AF-980907), the Swedish Brain Foundation (FO2021-0293), The Parkinson foundation of Sweden (1412/22), the Cure Alzheimer's fund, the Konung Gustaf V:s och Drottning Victorias Frimurarestiftelse, the Skåne University Hospital Foundation (2020-O000028), Regionalt Forskningsstöd (2022-1259) and the Swedish federal government under the ALF agreement (2022-Projekt0080). The Brain and Body Donation Program is supported by the National Institute of Neurological Disorders and Stroke (U24 NS072026 National Brain and Tissue Resource for Parkinson's Disease and Related Disorders), the National Institute on Aging (P30 AG19610 and P30 AG072980, Arizona Alzheimer's Disease Core Center), the Arizona Department of Health Services (contract 211002, Arizona Alzheimer's Research Center), the Arizona Biomedical Research Commission (contracts 4001, 0011, 05-901 and 1001 to the Arizona Parkinson's Disease Consortium) and the Michael J. Fox Foundation for Parkinson's Research. J.W.V. was supported by the SciLifeLab & Wallenberg Data Driven Life Science Program (grant: KAW 2020.0239). A.L.Y. was supported by a Skills Development Fellowship (MR/T027800/1) from the Medical Research Council and a Career Development Award from the Wellcome Trust [227341/Z/23/Z]. This research was funded in whole, or in part, by the Wellcome Trust [227341/Z/23/Z]. For the purpose of open access, the author has applied a CC BY public copyright licence to any Author Accepted Manuscript version arising from this submission. We would like to thank Per Borghammer for valuable discussions and recommendations throughout this study.

## Author contributions

S.E.M., O.H., J.W.V., R.O., L.E.C., and F.B. designed the study. S.E.M. performed the analyses and data interpretations under the supervision of O.H., J.W.V., R.O., L.E.C., F.B., and A.L.Y. The manuscript was drafted by S.E.M. All authors contributed to preparation and critical review of the manuscript. T.G.B., C.H.A., G.E.S., C.T., R.A.A., H.A.S., E.D.D., S.H.M., C.M.B., A.A., and P.C. collected the data.

## Funding

## Competing interests

L.E.C. has received research support from GE Healthcare (paid to institution). F.B. acts as a consultant for Biogen-Idec, IXICO, Merck-Serono, Novartis, Combinostics, and Roche. He has received grants, or grants are pending, from the Amyloid Imaging to Prevent Alzheimer's Disease (AMYPAD) initiative, the Biomedical Research Centre at University College London Hospitals, the Dutch MS Society, ECTRIMS–MAGNIMS, EU-H2020, the Dutch Research Council (NWO), the UK MS Society, and the National Institute for Health Research, University College London. He has received payments for the development of educational presentations from Ixico and his institution from Biogen-Idec and Merck. He is on the editorial board of Radiology, European Neuroradiology, Multiple Sclerosis Journal, and Neurology. Is on the board of directors of Queen Square Analytics. R.O. has received research support from Avid Radiopharmaceuticals, has given lectures in symposia sponsored by GE Healthcare and is an editorial board member of Alzheimer's Research & Therapy and the European Journal of Nuclear Medicine and Molecular Imaging. O.H. has acquired research support (for the institution) from ADx, AVID Radiopharmaceuticals, Biogen, Eli Lilly, Eisai, Fujirebio, GE Healthcare, Pfizer, and Roche. In the past 2 years, he has received consultancy/speaker fees from AC Immune, Amylyx, Alzpath, BioArctic, Biogen, Cerveau, Eisai, Eli Lilly, Fujirebio, Merck, Novartis, Novo Nordisk, Roche, Sanofi and Siemens. T.G.B. is a consultant for Aprinoia Therapeutics, Biogen and Avid Radiopharmaceuticals. A.A. has received, over the last 10 years, honoraria or support for consulting; participating in independent data safety monitoring boards; providing educational lectures, programs, and materials; or serving on advisory or oversight boards for AbbVie, Acadia, Allergan, the Alzheimer's Association, Alzheimer's Disease International, Axovant, AZ Therapies, Biogen, Eisai, Grifols, Harvard Medical School Graduate Continuing Education, JOMDD, Lundbeck, Merck, Prothena, Roche/Genentech, Novo Nordisk, Qynapse, Sunovion, Suven, and Synexus. Dr. Atri receives book royalties from Oxford University Press for a medical book on dementia. Dr. Atri receives institutional research grant/contract funding from NIA/NIH 1P30AG072980, NIA/NIH U22AG057437, AZ DHS CTR040636, Washington University St Louis, and Gates Ventures. Dr. Atri's institution receives/received funding for clinical trial grants, contracts, and projects from government, consortia, foundations and companies for which he serves/served as contracted site-PI. S.H.M. serves as a site co-investigator and receives funding for the PPMI study from the Michael J Fox Foundation. S.H.M. has received research funding from the Arizona Biomedical Research Consortium (ABRC) and International Essential Tremor Foundation (IETF). S.H.M. also has received funding for clinical trial grants, contracts, and projects from government and companies for which S.H.M. serves or has served as contracted site-PI. P.C. has received research support from Lewy Body Dementia association and Arizona Alzheimer's Consortium (both paid to institution). The remaining authors declare no competing interests.
