## [Peer Review File · Nature Communications]

Disease progression modelling reveals heterogeneity in trajectories of Lewy-type α -synuclein pathologyREVIEWER COMMENTS

Reviewer #1 (Remarks to the Author):

Mastenbroek et al. examined the heterogeneity in the spatial presentation of alpha-synuclein pathology (as quantified by Lewy-type alpha-synuclein density) across 10 brain regions using autopsy data from 814 individuals who had evidence of such pathology. They found three subtypes: the most common subtype (60.8% of individuals) exhibited earliest pathology in the olfactory bulb/tract and limbic areas. The other two subtypes were split about evenly between an olfactory and brainstem-early presentation and a brainstem-early but olfactory-later presentation. The authors' findings about spatial heterogeneity of alpha-synuclein pathology are consistent with the Unified Staging System for Lewy Body Disorders (USSLB), with the caveat that this system was based on the same cohort used in this study. The authors demonstrated differences in the clinicopathological diagnosis percentages across the three subgroups, other neuropathologies (amyloid and tau), APOE e4 carrier frequency, and motor and cognitive symptoms. Overall, their findings indicate that the most common subtype identified may mainly reflect cases with Alzheimer's disease and co-occurring Lewy body pathology. Additionally, the results are consistent with the idea that Parkinson's disease and dementia with Lewy bodies are subtypes within an alpha-synuclein pathology spectrum rather than separate entities. The authors also examined spinal cord and peripheral alpha-synuclein in a post-analysis and found that subtypes with early brainstem pathology had early and substantial peripheral pathology as well.

This paper makes important contributions to the literature by bringing a data-driven perspective for understanding the progression of alpha-synuclein pathology and helping reconcile the differences across the various (hypothetical) staging models.

It was a pleasure to read this excellent paper. The statistical analyses are of the highest quality. The authors have explored the data thoroughly and conducted meaningful sensitivity analyses to examine possible limitations. My only comment is a minor one: In the discussion of study limitations, it would be helpful to remind readers that these findings were based on cross-sectional data, and that longitudinal validation will be needed once in vivo datasets of alpha-synuclein (with longitudinal PET imaging) are available.

Reviewer #2 (Remarks to the Author):

This is a large-scale autopsy staging study from >800 patients with LBD pathology using new informatic approaches to define stages and subtypes of LBD pathology. The main findings are three subtypes defined by S1) OBT-Limbic distribution of LBD, S2) OBT-Brainstem distribution and S3) a relative sparing of OBT-brainstem group with relative distinct clinicopathologic features including greater AD co-pathology/APOE4 and cognitive impairment in S1 vs other groups, while S3 had greater PIGD and motor impairments as well as lower smell scores despite relative sparing of OBT. These data largely recapitulate the USBB LBD staging scheme with a new S3 group that has brainstem relative pathology with mild OBT involvement and in the subgroup with peripheral tissue and Spinal cord data, greater Lewy pathology in these tissues in S2/3 vs S1, supportive of an early-extra CNS involvement in a subset of patients, similar to recent "body-first" LBD subgroup proposed. The authors conclude that clinical heterogeneity of LBD can be inferred from subgroups with different spatial patterns of LBD progression.

This autopsy data is important and potentially impactful and there are several strengths to the manuscript including the large community based autopsy cohort, rigor of the neuropathology methods (including rigor of evaluation of the olfactory bulb), rigor of the statistical analyses with sensitivity analysis to account for missing data. While the depth of the neuropathology data and unique statistical modeling is impressive, enthusiasm is somewhat weakened by the lack of clear novelty of the findings as the results appear to recapitulate previous data from this cohort such as the USSBL staging scheme and other informatic analyses of this and other cohorts (e.g. Toledo et al, *Acta Neuropathologica* 2016). It would be helpful for the authors to clearly articulate what new knowledge is gained from these detailed analyses and specifically how many cases overlap with the original USSBL study. Moreover, the data for differentiating the S3 subgroup from the S2 subgroup, which is perhaps the most novel finding, is relatively weak as both groups have mild olfactory LBD pathology in early stages (<5) and reach severe stages of pathology at roughly stage 30 in both groups (Figure 3, Supplementary Figures S2,3). Due to the methodological issues nicely and rigorously addressed by the authors by re-staining the negative cases, it is still not clear that ordinal ratings of moderate to severe are reproducible and accurate enough to substantiate S3 as a separate group from S2, as between group differences of regional pathology are relatively modest (Figure 3). It is also counter-intuitive that the S3 group has worse olfaction despite relative sparing. Thus, the claims of S3 being affected later in the disease in S3 do not appear fully supported by the data since it appears most have at least mild pathology in stages <5. The data appears to represent more of a spectrum of PD vs PDD in S3 vs S2 than a discrete pathological subgroup in the current analysis.

It would also be helpful for the authors to use modern nomenclature and clinical (i.e. McKeith et al 2017, Emre et al 2016, etc.) and pathological criteria (Montine et al 2011- including ABC scores for all patients regardless of clinical phenotype) to classify their patients rather than the clinicopathological diagnoses included which are not clear. For example, for those classified as AD it is not clear if this is due to plaque and tangle scores or by a multidomain amnesic syndrome- it is not clear based on their staging if a patient who met clinical criteria for DLB but had high AD co-pathology would be classified as AD.

Finally the claim ‘Taken together, our results suggest that the S1 (OBT-early/Limbic-early) subtype might primarily reflect AD cases with co-occurring LB pathology.’ (page 17, line 343-44) is not clearly supported by the data as their analyses did not examine potential stages of tau or amyloid beta. Additionally, more guarded language on the interpretation of the body Lewy pathology data for S3 could be used due to the low N of this subgroup for peripheral tissue scores.

I have a few additional minor comments below:

It would be helpful for the authors to account for the n in each epoch of their sliding window analysis in Supplemental figure 7.

If available it would be helpful to have GBA mutation status data, as this is an important genetic driver of relatively pure neocortical LBD. If available it would also be helpful to include data on nigral loss to support the brainstem sparing claims of S1 as in the USSLB study.

We thank the Editor and reviewers for taking the time to review our manuscript and providing constructive feedback to improve our manuscript. We have revised the manuscript accordingly. Please find below the original comments from reviewers in black and our corresponding responses in blue.

REVIEWER COMMENTS

Reviewer #1 (Remarks to the Author):

Mastenbroek et al. examined the heterogeneity in the spatial presentation of alpha-synuclein pathology (as quantified by Lewy-type alpha-synuclein density) across 10 brain regions using autopsy data from 814 individuals who had evidence of such pathology. They found three subtypes: the most common subtype (60.8% of individuals) exhibited earliest pathology in the olfactory bulb/tract and limbic areas. The other two subtypes were split about evenly between an olfactory and brainstem-early presentation and a brainstem-early but olfactory-later presentation. The authors' findings about spatial heterogeneity of alpha-synuclein pathology are consistent with the Unified Staging System for Lewy Body Disorders (USSLB), with the caveat that this system was based on the same cohort used in this study. The authors demonstrated differences in the clinicopathological diagnosis percentages across the three subgroups, other neuropathologies (amyloid and tau), APOE e4 carrier frequency, and motor and cognitive symptoms. Overall, their findings indicate that the most common subtype identified may mainly reflect cases with Alzheimer's disease and co-occurring Lewy body pathology. Additionally, the results are consistent with the idea that Parkinson's disease and dementia with Lewy bodies are subtypes within an alpha-synuclein pathology spectrum rather than separate entities. The authors also examined spinal cord and peripheral alpha-synuclein in a post-analysis and found that subtypes with early brainstem pathology had early and substantial peripheral pathology as well.

This paper makes important contributions to the literature by bringing a data-driven perspective for understanding the progression of alpha-synuclein pathology and helping reconcile the differences across the various (hypothetical) staging models.

It was a pleasure to read this excellent paper. The statistical analyses are of the highest quality. The authors have explored the data thoroughly and conducted meaningful sensitivity analyses to examine possible limitations. My only comment is a minor one: In the discussion of study limitations, it would be helpful to remind readers that these findings were based on cross-sectional data, and that longitudinal validation will be needed once in vivo datasets of alpha-synuclein (with longitudinal PET imaging) are available.

Authors' response: We thank the reviewer for their appreciation of our work and their suggested improvement.

We have added the cross-sectional *postmortem* study design as a limitation to the discussion, page 20:

“Finally, the cross-sectional study design warrants future validation in longitudinal *in vivo* datasets, once α -synuclein Positron Emission Tomography (PET) imaging is available¹.”

1. Smith, R. et al. The α -synuclein PET tracer [18F] ACI-12589 distinguishes multiple system atrophy from other neurodegenerative diseases. *Nature Communications* 14, 6750 (2023). <https://doi.org/10.1038/s41467-023-42305-3>

Reviewer #2 (Remarks to the Author):

This is a large-scale autopsy staging study from >800 patients with LBD pathology using new informatic approaches to define stages and subtypes of LBD pathology. The main findings are three subtypes defined by S1) OBT-Limbic distribution of LBD, S2) OBT-Brainstem distribution and S3) a relative sparing of OBT-brainstem group with relative distinct clinicopathologic features including greater AD co-pathology/APOE4 and cognitive impairment in S1 vs other groups, while S3 had greater PIGD and motor impairments as well as lower smell scores despite relative sparing of OBT. These data largely recapitulate the USSBL LBD staging scheme with a new S3 group that has brainstem relative pathology with mild OBT involvement and in the subgroup with peripheral tissue and Spinal cord data, greater Lewy pathology in these tissues in S2/3 vs S1, supportive of an early-extra CNS involvement in a subset of patients, similar to recent "body-first" LBD subgroup proposed. The authors conclude that clinical heterogeneity of LBD can be inferred from subgroups with different spatial patterns of LBD progression.

Authors' response: We thank the reviewer for their interest in our work and their suggested improvements. Below, please find a detailed response to each comment.

This autopsy data is important and potentially impactful and there are several strengths to the manuscript including the large community based autopsy cohort, rigor of the neuropathology methods (including rigor of evaluation of the olfactory bulb), rigor of the statistical analyses with sensitivity analysis to account for missing data. While the depth of the neuropathology data and unique statistical modeling is impressive, enthusiasm is somewhat weakened by the lack of clear novelty of the findings as the results appear to recapitulate previous data from this cohort such as the USSBL staging scheme and other informatic analyses of this and other cohorts (e.g. Toledo et al, *Acta Neuropathologica* 2016). It would be helpful for the authors to clearly articulate what new knowledge is gained from these detailed analyses and specifically how many cases overlap with the original USSBL study.

Authors' response: Regarding the novelty of the current work, we recognize that patterns similar to the S1: OBT-early/Limbic-early and S2: OBT-early/Brainstem-early patterns of Lewy body (LB) pathology have been described previously in the current cohort, i.e. in the original USSBL study. Nevertheless, we believe that the current study adds to the pre-existing literature in several ways.

First, while a subset of the included cases overlaps with the original USSBL study ($n=267$ [32.8%]), we expanded the dataset significantly with 814 as compared to 417 cases in the

original study. This unique, large, and extensively characterized *postmortem* dataset enabled us to investigate the existence of multiple LB progression patterns with greater power.

Second, the large sample size enabled us to adopt a different staging approach compared to the original USSLB study: where the USSLB study constructed a staging scheme using a qualitative regional analysis, the current study adopted a data-driven modelling approach, which allows for a more detailed description of LB trajectories. Here, we extend on the previously defined USSLB by incorporating a larger number of brain regions, information on the extent of pathology, and a pseudo-temporal component. In addition, the current study allows for more in-depth individual staging; by not only estimating the current stage of the individual, but also the most probable starting point of pathology. This is especially important for cases with advanced disease and widespread pathology as is often the case in *postmortem* studies. Where the USSLB speculates on the origin of pathology in such cases, the current study provides a statistical framework suggesting how pathology most likely got to this endpoint. As a result, we could characterize and compare the different LB progression patterns in a more comprehensive manner, spanning the whole range of pathology from very early to late-stage pathology, whereas the USSLB-predicted pathways were based only on the patterns observed in stage IIa (brainstem-predominant) and IIb (limbic-predominant) cases.

As the reviewer points out, Toledo et al. (2016) has previously investigated LB patterns in a data-driven manner in, among others, the Banner cohort. However, there are several important differences between Toledo et al. and the current study. First, where Toledo et al. focused on autopsy cases diagnosed *postmortem* with PD, DLB, or (concurrent) AD, we included all donors with any evidence of brain LB pathology, resulting in a larger sample size ($n=542$ vs $n=814$) and a broader range of disease stages, including those with earliest pathology (i.e., incidental Lewy body disease or other neurodegenerative diseases with concurrent LB pathology). Enrichment for early-stage pathology is especially important when investigating the origin of pathology in a *postmortem* dataset, where the majority would be late-stage with widespread pathology. Second, Toledo et al. used a different modelling approach, exclusively focusing on clusters of cases with similar spatial patterns of pathology, while disregarding the temporal evolution of LB pathology. Hence, where the study of Toledo et al. presents five clusters that likely present different disease stages that succeed each other over the disease course, the current study inferred three trajectories spanning the entire disease course. In addition, it has been shown previously that SuStaIn can capture more information and variability than is possible for stage-only models (Vogel et al., 2021, *Nature Medicine*; Young et al., 2023, *Brain*).

Finally, one of the main differences of the current analysis, compared with both the original USSLB and Toledo et al. (2016) studies, is the identification of a third, novel, regional progression pattern, characterized by early brainstem pathology, as described in more detail below.

We have added a brief statement on novelty of the work to the Discussion, page 21: “To conclude, by applying a data-driven modelling approach to postmortem density scores assessed in a large number of regions, we show significant heterogeneity in LB spreading trajectories, supporting and extending on previous literature. Specifically, in line with previous studies, we identified two subtypes that show earliest pathology in the olfactory bulb followed by either limbic or brainstem regions. In addition, we describe a novel subtype with initial quantitatively greater involvement of the brainstem, reaching severe levels of olfactory bulb pathology relatively later.”

Moreover, the data for differentiating the S3 subgroup from the S2 subgroup, which is perhaps the most novel finding, is relatively weak as both groups have mild olfactory LBD pathology in early stages (<5) and reach severe stages of pathology at roughly stage 30 in both groups (Figure 3, Supplementary Figures S2,3). Due to the methodological issues nicely and rigorously addressed by the authors by re-staining the negative cases, it is still not clear that ordinal ratings of moderate to severe are reproducible and accurate enough to substantiate S3 as a separate group from S2, as between group differences of regional pathology are relatively modest (Figure 3). It is also counter-intuitive that the S3 group has worse olfaction despite relative sparing. Thus, the claims of S3 being affected later in the disease in S3 do not appear fully supported by the data since it appears most have at least mild pathology in stages <5. The data appears to represent more of a spectrum of PD vs PDD in S3 vs S2 than a discrete pathological subgroup in the current analysis.

Authors' response: As the reviewer points out, one of the main pathological differences between the S2 (OBT-early/Brainstem-early) and S3 (Brainstem-early/OBT-later) subtypes is the temporal development of OBT pathology. We would like to point out that while both S2 and S3 develop mild OBT pathology (density score=1) in early stages (S2 stage 1; S3 stage 4), S2 develops severe pathology (density score =3) in stage 3, whereas S3 reaches severe pathology in stage 27. Hence, the differences in OBT involvement are not based on mild compared to moderate, but mild compared to severe pathology (score=1 vs score=3). We would like to argue that a difference between mild and severe pathology and a timing of stage 3 vs stage 27 is quite substantial. We do, however, realize the limitations of ordinal pathology ratings, and while SuStaIn takes this uncertainty into account by working with probabilities rather than absolute ordinal values (Supplementary Table 4), future studies would be needed to look at continuous load, rather than ordinal scores, by adopting a more quantitative molecular measure.

Besides the evolution of OBT pathology, S2 and S3 show additional differences in LB pathology. First, the positional variance diagrams in Figure 2 show that, in S3, besides the OBT, many of the other brain regions reach mild to moderate pathology almost concurrently, suggesting rapid spreading throughout the brain. Severe and very severe levels of pathology are only reached in later stages of S3. In contrast, S2 (and S1) subtypes seem to only sequentially accumulate extensive pathology, i.e. severe or very severe, in one region before another region is affected, suggesting slower spread. This is further illustrated by the figure below, showing the number of brain regions to have any pathology (density score > 0; left) and severe to very severe pathology (density score > 2; right), while in early SuStaIn stages (<20). While the S3 (Brainstem-early/OBT-later) early-SuStaIn cases clearly have on average more brain regions with non-zero pathology compared to the other subtypes, the S3 regional pathology loads are also lighter. With regards to Reviewer Comment 1, this observation could be considered a novel finding as compared to previous studies.

We have added this finding to the Results, section Three heterogeneous disease progression patterns of Lewy-type α -synucleinopathies, page 9:

“Compared to the other subtypes, individuals in the S3 (Brainstem-early/OBT-later) subtype seem to develop mild LB pathology across most brain regions early on, suggesting rapid spreading throughout the brain (**Figure 2**). In contrast, both the S1 (OBT-early/Limbic-early) and S2 (OBT-early/Brainstem-early) seem to sequentially accumulate extensive pathology, affecting one region before another is affected. This is further illustrated by the finding that, in early disease stages (SuStaIn stage<20), individuals assigned to the S3 (Brainstem-early/OBT-later) subtype have on average a larger number of brain regions that show non-zero LTS, and fewer regions that show severe or very severe LTS (**Figure 3**).”

Second, S3 (Brainstem-early/OBT-later) has different regional patterns of LB pathology as compared to S2 (OBT-early/Brainstem-early) (Figure 3). Specifically, S3 has higher levels of temporal, cingulate, and entorhinal LB pathology, whereas S2 has more OBT pathology. We would like to point out that most of these differences are not modest, but actually reach a t-value between -5 and -10, indicating strong effects.

More support for S2 and S3 being distinct LB trajectories are the observed differences in clinical profiles across subtypes, specifically regarding LBD-related symptoms (i.e., hyposmia and motor symptoms). To rule out that these differences were observed by chance, we split the data into three random groups 1000 times, each time ensuring with sample sizes and total pathology levels matched the three identified subtypes (n=409; n=131; n=120). Subsequently, for each random split, we performed the statistically compared hyposmia and motor symptoms between the two smaller groups (resembling S2 and S3). We used these 1000 comparisons as a null distribution, which we used to calculate the probability of finding a t-value higher than the t-value observed in the comparison of S2 vs S3 by chance given the data (t motor symptoms = 2.634; t hyposmia= 2.996) (number of higher t-values/1000). We observed a probability of p=0.002 and p=0.013 for motor and smell symptoms, respectively. This suggests that SuStaIn used patterns of pathology to group individuals into subgroups with functionally relevant differences in a manner that exceeded chance. This suggests that the clinical differences between S2 and S3 are not merely observed by chance, but most likely represent real differences between pathological subtypes.

Regarding the worse olfaction in S3 despite relative sparing of the OBT, it has been reported previously that total brain LB pathology is more strongly correlated to olfactory function than OBT pathology, with OBT pathology not being related to olfactory function unless pathology in other brain regions is also observed (Tremblay et al., 2022, Brain Pathol). This is also described in the Discussion, page 19: “Interestingly, despite relatively late involvement of olfactory regions, the S3 (Brainstem-early/OBT-later) subtype exhibited worse smell impairment compared to both other subtypes. Of note, previous studies have shown that it is not the extent of olfactory bulb LB pathology, but rather the total brain LB pathology, that is associated with smell function^{53,54}, for which subtype comparisons in this study were adjusted.” Therefore, while somewhat unintuitive, our finding is not inconsistent with previous reports.

It would also be helpful for the authors to use modern nomenclature and clinical (i.e. McKeith et al 2017, Emre et al 2016, etc.) and pathological criteria (Montine et al 2011- including ABC scores for all patients regardless of clinical phenotype) to classify their patients rather than the clinicopathological diagnoses included which are not clear. For example, for those classified as AD it is not clear if this is due to plaque and tangle scores or by a multidomain amnesic syndrome- it is not clear based on their staging if a patient who met clinical criteria for DLB but had high AD co-pathology would be classified as AD.

Authors’ response: As stated in the Methods section of this paper, the diagnostic criteria used were those published in the Beach et al. AZSAND paper, 2015¹. These criteria are based on international consensus criteria that use both clinical and neuropathological findings, and so are termed "clinicopathological". As the cases were autopsied between 1997 and 2021, diagnostic criteria for both AD and DLB changed over that time span, including, as the reviewer mentions, the additions of the 2011 NIA-AA AD criteria (Hyman et al, Montine et al 2011), and the McKeith et al 2017 DLB criteria. Therefore, we used single sets of diagnostic criteria that spanned this entire time period, i.e. NIA-Reagan (1991) criteria for diagnosing AD (with NIA-Reagan, intermediate and high Alzheimer’s disease neuropathological changes plus clinical dementia are sufficient for a diagnosis of AD regardless of co-pathologies) and McKeith et al (2005) criteria diagnostic criteria for the diagnosis of DLB (intermediate and high Lewy synucleinopathy plus clinical dementia are sufficient for a diagnosis of DLB regardless of co-pathologies). With hundreds of autopsy cases to reclassify, it is not possible at this moment to redo these classifications to NIA-AA and McKeith et al 2017 for all cases, due to the workload involved, although we are currently working through this task.

We note that a couple of publications have found not much difference in classification between the NIA-Reagan and NIA-AA criteria² and the McKeith et al 2017 neuropath criteria³ basically did not change from the 2005 criteria⁴.

For the possible differences in AD diagnosis using the NIA-Reagan vs NIA-AA criteria, we searched the current BBDP database for discrepant classifications. Of a total of 421 cases classified with both systems, there were only 36 discrepancies (8.5%). Of these, 32 were due to differences between intermediate and high assignments and these would not affect the assignment of an AD diagnosis. Only 4 discrepancies (0.95%) were due to differences in the assignment of low or intermediate. Those assigned to "low" would not meet diagnostic criteria for AD.

In the current paper, subjects diagnosed as PD or DLB, that also met the diagnostic criteria for AD were classified as mixed AD/PD and mixed AD/DLB.

1. Beach, T. G., et al. (2015). "Arizona Study of Aging and Neurodegenerative Disorders and Brain and Body Donation Program." *Neuropathology* 35(4): 354-389.
2. Serrano-Pozo A, et al (2016). Thal Amyloid Stages Do Not Significantly Impact the Correlation Between Neuropathological Change and Cognition in the Alzheimer Disease Continuum. *J Neuropathol Exp Neurol*. 2016 Jun;75(6):516-26. doi: 10.1093/jnen/nlw026. Epub 2016 Apr 22. PMID: 27105663; PMCID: PMC6250207.
3. McKeith IG, et al. (2017). Diagnosis and management of dementia with Lewy bodies: Fourth consensus report of the DLB Consortium. *Neurology*. 2017 Jul 4;89(1):88-100. doi: 10.1212/WNL.0000000000004058. Epub 2017 Jun 7. PMID: 28592453; PMCID: PMC5496518.
4. McKeith IG, et al. (2005). Consortium on DLB. Diagnosis and management of dementia with Lewy bodies: third report of the DLB Consortium. *Neurology*. 2005 Dec 27;65(12):1863-72. doi: 10.1212/01.wnl.0000187889.17253.b1. Epub 2005 Oct 19. Erratum in: *Neurology*. 2005 Dec 27;65(12):1992. PMID: 16237129.

Finally the claim ‘Taken together, our results suggest that the S1 (OBT-early/Limbic-early) subtype might primarily reflect AD cases with co-occurring LB pathology.’ (page 17, line 343-44) is not clearly supported by the data as their analyses did not examine potential stages of tau or amyloid beta. Additionally, more guarded language on the interpretation of the body Lewy pathology data for S3 could be used due to the low N of this subgroup for peripheral tissue scores.

Authors’ response: We would like to point out that we did analyse differences in total amyloid plaque and tau tangle burden, Results section, page 13:

“The S1 (OBT-early/Limbic-early) subtype consisted of more *APOE* ϵ 4 carriers (vs S2: $\beta=0.40$, $p=0.035$; vs S3: $\beta=0.68$, $p=0.001$; **Figure 6B**) and had more plaque and neurofibrillary pathology than the other two subtypes (vs S2: $\beta_{\text{plaques}}=2.20$, $p_{\text{plaques}}<0.001$, $\beta_{\text{neurofibrillary pathology}}=2.65$, $p_{\text{neurofibrillary pathology}}<0.001$; vs S3: $\beta_{\text{plaques}}=4.77$, $p_{\text{plaques}}<0.001$, $\beta_{\text{neurofibrillary pathology}}=4.12$, $p_{\text{neurofibrillary pathology}}<0.001$; **Figure 6C-D**).”

We observed more tau and amyloid pathology in the S1 (OBT-early/Limbic-early) subtype as compared to both S2 and S3. Nevertheless, we have reformulated the statement on page 17: “Taken together, our results suggest that the S1 (OBT-early/Limbic-early) subtype might primarily reflect cases with co-occurring AD and LB pathology.”

In addition, we have rewritten the interpretation of the Lewy body pathology data throughout the discussion, highlighting the small sample size of the S3 subtype. Discussion, page 16: “Finally, supporting prior observations²⁵, a comparison of non-brain LB pathology across subtypes showed that individuals with early pathology in brainstem regions exhibited substantial LB pathology in the spinal cord and peripheral regions from the earliest disease stages, with a tendency towards more non-brain pathology in those subjects where the brainstem is the first region to be affected, although it should be noted that few datapoints were available in this subtype.”

I have a few additional minor comments below:

It would be helpful for the authors to account for the n in each epoch of their sliding window analysis in Supplemental figure 7.

Authors' response: We have added the sample size of each Lewy body subtype in each window in Supplemental figure 7 (now Supplemental figure 9), as can be appreciated below. The subtype sizes range from $n=56-150$ for S1, $n=20-49$ for S2, and $n=15-49$ for S3.

Subtype sizes for each window

Subtype	1-10	2-11	3-12	4-13	5-14	6-15	7-16	8-17	9-18	10-19	11-20	12-21	13-22	14-23	15-24	16-25	17-26	18-27	19-28	20-29	21-30	22-31	23-32	24-33	25-34	26-35	27-36	28-37	29-38	30-39	31-40
S1	150	139	132	108	95	93	85	79	82	78	69	62	63	56	57	59	61	60	67	83	98	108	113	113	114	117	113	108	91	83	80
S2	32	35	42	46	44	39	38	45	42	44	45	45	40	45	55	49	48	44	41	41	40	32	20	22	25	25	31	32	31	30	28
S3	36	30	31	26	27	27	25	25	22	19	20	29	29	36	43	43	45	48	49	38	40	32	27	26	24	18	23	25	19	17	15

If available it would be helpful to have GBA mutation status data, as this is an important genetic driver of relatively pure neocortical LBD. If available it would also be helpful to include data on nigral loss to support the brainstem sparing claims of S1 as in the USSLB study.

Authors' response: We thank the reviewer for this interesting suggestion.

We had 368 cases with data on GBA mutation status. Only a few cases showed a positive test result, as can be appreciated in the figure below. There were no statistically significant differences between the subtypes (logistic regression adjusted for age, sex, and SuStaIn stage), although it seems that GBA mutation carriers were more frequent in S2 and S3, as compared to S1.

In addition, we compared the Lewy body subtypes on substantia nigra (SN) pigmented neuron loss, graded as none, mild, moderate or severe. We used an ordinal regression model, adjusted for age, sex, and SuStaln stage. As can be appreciated below, individuals in the S3: Brainstem-early/OBT-later subtype showed on average more nigral loss compared to the S1: OBT-early/Limbic-early subtype (SN level 2: $\beta=3.10$, $p=0.001$; SN level 3: $\beta=4.35$; $p<0.001$) and the S2: OBT-early/Brainstem-early (SN level 3: $\beta=3.20$ $p=0.003$). There was no significant difference between S1 and S2. These results highlight another phenotypic difference between S2 and S3, despite accounting for age, sex, and disease stage.

We have added these two additional comparisons to the manuscript page 13 and Supplementary Figure 7:

“In addition, the S3 (Brainstem-early/OBT-later) subtype showed more severe substantia nigra depigmentation as compared to the other two subtypes (vs S1: SN level 2: $\beta=3.10$, $p=0.001$, SN level 3: $\beta=4.35$; $p<0.001$; vs S2: SN level 3: $\beta=3.20$ $p=0.003$; **Figure S7**). No differences in GBA mutation status were observed (**Figure S7**).”

REVIEWERS' COMMENTS

Reviewer #1 (Remarks to the Author):

I'd like to congratulate the authors on their thorough responses to reviewer comments and their excellent revision. They addressed my comment and I have no additional suggestions.

Reviewer #2 (Remarks to the Author):

The authors have been very responsive to the previous round of reviews including a more clear articulation of the novelty of the study and additional analyses to demonstrate additional pathological differences between S2 and S3 groups, including new data with SN loss. The rationale for lack of modern clinical and pathological criteria are justified by the large amount of work to modernize from legacy cases. Minor optional recommendations include, if room including a figure to show differences in OFB ratings to help readers understand the magnitude of mild to severe scores that differentiate the two groups, some measure of inter-rater reliability of OFB scores and mention of limitation of clinical and pathological criteria used.

We thank the Editor and reviewers for taking the time to review our manuscript and providing constructive feedback to improve our manuscript. We have revised the manuscript accordingly. Please find below the original comments from reviewers in black and our corresponding responses in blue.

REVIEWERS' COMMENTS

Reviewer #2 (Remarks to the Author):

The authors have been very responsive to the previous round of reviews including a more clear articulation of the novelty of the study and additional analyses to demonstrate additional pathological differences between S2 and S3 groups, including new data with SN loss. The rationale for lack of modern clinical and pathological criteria are justified by the large amount of work to modernize from legacy cases. Minor optional recommendations include, if room including a figure to show differences in OFB ratings to help readers understand the magnitude of mild to severe scores that differentiate the two groups, some measure of inter-rater reliability of OFB scores and mention of limitation of clinical and pathological criteria used.

Author's response: We thank the reviewer for their suggested improvements. We have revised the manuscript accordingly.

We have added a Supplementary Figure showing example pictures of LB density ratings in the olfactory bulb:

“Figure S10. Example images of various LTS density scores in the olfactory bulb

Photomicrographs of the immunohistochemical staining for α -synuclein in the olfactory bulb. Positive immunostaining is shown in black; the counterstain is Neutral Red. **A.** Mild pathology. **B.** Moderate pathology. **C.** Severe pathology. **D.** Very severe pathology.”

Regarding the inter-rater reliability, we have previously obtained a high correlation of 0.85 between independent ratings of olfactory bulb density scores (unpublished data in preparation for Tremblay et al.^{1,2}). However, the study of Attems et al.³ reported a moderate inter-rater correlation of ≈ 0.67 for olfactory bulb semi-quantitative scoring, across several staging methods. Of note, these observers were not specifically trained in the grading system used in the grading systems used in Tremblay et al. Nonetheless, we have referenced the study of Attems et al. in the Methods, page 22.

We have added the limitation of the clinico-pathological criteria used to the discussion, page 20: “Third, as the cases were autopsied between 1997 and 2021, AD and DLB clinicopathological diagnoses are based on older iterations of neuropathological criteria, although changes in criteria are minor [4,25,69].

- 1 Tremblay, C. *et al.* Olfactory Bulb Amyloid- β Correlates With Brain Thal Amyloid Phase and Severity of Cognitive Impairment. *J Neuropathol Exp Neurol* **81**, 643-649 (2022). <https://doi.org:10.1093/jnen/nlac042>
- 2 Tremblay, C. *et al.* Effect of olfactory bulb pathology on olfactory function in normal aging. *Brain Pathology* **32**, e13075 (2022). <https://doi.org:https://doi.org/10.1111/bpa.13075>
- 3 Attems, J. *et al.* Neuropathological consensus criteria for the evaluation of Lewy pathology in post-mortem brains: a multi-centre study. *Acta Neuropathologica* **141**, 159-172 (2021). <https://doi.org:10.1007/s00401-020-02255-2>